# Drugging the Undruggable: Benchmarking and Modeling Fragment-Based Screening

**Haichuan Tan**[1,2][*]**, Bowen Gao**[1,2][*]**, Jiaxin Li**[3]**, Yinjun Jia**[1]**, Wenyu Zhu**[1]**,**
**Wenxuan Xie**[4]**, Yihong Liu**[2]**, Yanwen Huang**[5]**, Jianhui Wang**[1]**, Yuanhuan Mo**[6]**,**
**Ya-Qin Zhang**[1]**, Wei-Ying Ma**[1]**, Yanyan Lan**[1,7][†]

[1]Institute for AI Industry Research (AIR), Tsinghua University
[2]Department of Computer Science and Technology, Tsinghua University
[3]Beijing University of Posts and Telecommunications
[4]Fudan University
[5]Department of Pharmaceutical Science, Peking University
[6]South China University of Technology
[7]Beijing Frontier Research Center for Biological Structure, Tsinghua University

## Abstract

A significant portion of disease-relevant proteins remain undruggable due to shallow, flexible, or otherwise ill-defined binding pockets that hinder conventional molecule screening. Fragment-based drug discovery (FBDD) offers a promising alternative, as small, low-complexity fragments can flexibly engage shallow, transient, or cryptic binding pockets that are often inaccessible to conventional drug-like molecules. However, fragment screening remains difficult due to weak binding signals, limited experimental throughput, and a lack of computational tools tailored for this setting. In this work, we introduce **FragBench**, the first benchmark for fragment-level virtual screening on undruggable targets. We construct a high-quality dataset through multi-agent LLM–human collaboration and interaction-based fragment labeling. To address the core modeling challenge, we propose a novel tri-modal contrastive learning framework **FragCLIP** that jointly encodes fragments, full molecules, and protein pockets. Our method significantly outperforms baselines like docking software and other ML based methods. Moreover, we demonstrate that retrieved fragments can be effectively expanded or linked into larger compounds with improved predicted binding affinity, supporting their utility as viable starting points for drug design.

## 1 Introduction

A substantial fraction of disease-associated proteins are considered undruggable targets, such as transcription factors (Bushweller, 2019) and protein–protein interaction (PPI) hubs (Arkin & Wells, 2004). These proteins are closely linked to severe diseases, including cancer and neurodegenerative disorders (Bushweller, 2019; Ross & Poirier, 2004). If therapeutic strategies could be developed for them, the clinical and societal impact would be transformative. However, the lack of well-defined, stable binding pockets makes conventional approaches—small-molecule drug design and high-throughput screening—largely ineffective, thus severely limiting progress in drug discovery for these targets.

Fragment-based drug discovery (FBDD) offers a unique path forward for undruggable targets. Compared to drug-like molecules, fragments are smaller and more flexible in their binding modes, which enables them to access shallow or transient binding pockets on protein surfaces (Erlanson et al., 2016) and to reveal weak but crucial interactions often invisible to conventional screening. Although individual fragments bind weakly, they can serve as anchors that can be expanded or linked

---

[*]Equal contribution
[†]Correspondence to `lanyanyan@air.tsinghua.edu.cn`

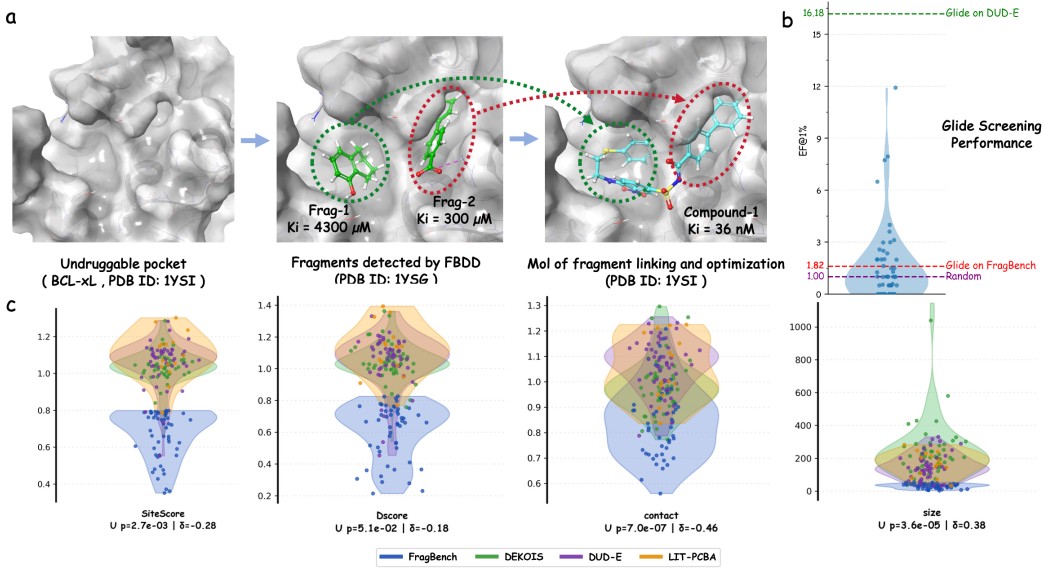

Figure 1: (a) Fragment-based drug discovery on BCL-xL, a shallow-pocket undruggable target where conventional screening fails. Weakly binding fragments identified by NMR were linked to yield compound-1 with markedly improved affinity. (b) Glide (Halgren et al., 2004) shows poor enrichment on such targets (EF1 = 1.8), barely above random. (c) Pocket property distributions of FragBench (green) versus DUD-E (purple), with significance tested by Mann–Whitney U and effect sizes reported as Cliff's $\delta$.

to yield high-affinity, selective molecules (Murray & Rees, 2009). This principle has been validated in classical undruggable targets, highlighting the promise of FBDD in this space (Scott et al., 2016).

Over 85% of the human proteome remains "undruggable" due to the absence of well-defined pockets suitable for conventional small-molecule targeting (Spradlin et al., 2021). Fragment-based drug discovery (FBDD) offers a promising alternative, exemplified by pipelines like Enamine's V-SYNTHES, which dock fragment-like synthons and expand them through iterative synthesis (Sadybekov et al., 2022). Figure 1(a) illustrates BCL-xL, a classical undruggable target, as an example of developing a full-molecule ligand through FBDD. However, the initial fragment identification step remains a bottleneck. Experimental techniques such as NMR and crystallography reliably detect weak binders but are slow, costly, and limited by factors like solubility or crystal quality (Erlanson et al., 2016; Jhoti et al., 2007). Computational docking, originally designed for full-sized ligands, often underestimates small fragments, leading to high false positive and negative rates (Brenke et al., 2009; Chen & Pohlhaus, 2010). As shown in Figure 1(b), our experiments confirm these limitations—docking achieves only marginally better fragment ranking than random. These challenges highlight the need for fragment-aware computational models tailored to FBDD and undruggable targets.

Despite the rapid advances of machine learning in drug discovery, AI-driven approaches for fragment screening remain largely unexplored. Two major gaps currently hinder progress in this area. First, there is a lack of systematic benchmarks specifically designed for fragment screening on undruggable protein pockets. This absence limits standardized evaluation and meaningful comparison between methods. Second, existing modeling frameworks fail to capture the tripartite relationship among fragments, drug-like molecules, and protein pockets. Without explicitly modeling both fine-grained fragment–pocket interactions and global molecule–pocket binding principles, it is difficult to achieve generalizable fragment retrieval across diverse targets.

To bridge the current gaps in fragment-based drug discovery, we introduce **FragBench**, the first large-scale benchmark specifically curated for fragment screening on undruggable targets. It features high-confidence positive and negative fragment–pocket pairs derived from structurally challenging protein complexes. Building upon this, we propose **FragCLIP**, a tri-modal contrastive learning

framework that jointly encodes fragments, molecules, and protein pockets, aligning their representations through a dedicated fusion module. This design captures both fine-grained fragment–pocket interactions and scaffold-level molecular context. Our experiments show that FRAGCLIP significantly outperforms classical docking tools and recent learning-based models—especially in the challenging cross-target setting—highlighting its potential to advance AI-driven fragment discovery for undruggable proteins.

## 2 RELATED WORK

Recent advances in deep learning have markedly improved structure-based virtual screening (VS). Models such as EquiBind and DiffDock (Stärk et al., 2022; Corso et al., 2022) enable fast and accurate prediction of protein–ligand binding conformations, while multimodal frameworks like DrugCLIP (Gao et al., 2023) employ contrastive learning to align protein and ligand representations for efficient screening. Meanwhile, standardized benchmarks such as DUD-E, LIT-PCBA, and CrossDocked2020 (Mysinger et al., 2012; Tran-Nguyen et al., 2020; Francoeur et al., 2020) have facilitated robust evaluation of virtual screening models. However, these efforts overwhelmingly focus on *drug-like molecules* and well-structured binding pockets, overlooking the unique challenges of *fragment-based screening*—where small, low-affinity fragments interact with shallow or transient sites that are typical of *undruggable* targets.

Fragment-based drug discovery (FBDD) offers a compelling strategy, as fragments can access cryptic sites and seed ligand development. However, efficient screening remains challenging: experimental methods like NMR and crystallography are accurate but low-throughput, while computational tools such as hotspot mapping and fragment docking often misrank fragments due to scoring biases. Despite progress in virtual screening (VS), no existing benchmark or framework systematically addresses fragment screening on undruggable targets or captures the interplay among fragments, molecules, and pockets.

## 3 METHOD

Fragment-based drug discovery (FBDD) holds promise for targeting undruggable proteins with shallow or cryptic pockets, where traditional small-molecule screening fails. However, the core task of fragment-level virtual screening remains underdefined and challenging: fragment–pocket binding signals are weak, supervision is scarce, and existing scoring or retrieval methods—typically optimized for drug-like ligands—fail to generalize to low-mass fragments.

To address this, we formalize the task of fragment retrieval in undruggable pockets, and introduce a tri-modal modeling approach that leverages information from protein pockets, full drug molecules, and their constituent fragments. Our solution includes (i) **FragBench**, a new benchmark built with LLM-guided literature mining and interaction-based fragment labeling; and (ii) **FragCLIP**, a contrastive framework that aligns fragment, molecule, and pocket embeddings via multi-level supervision.

### 3.1 FRAGBENCH: FRAGMENT-BASED BENCHMARK FOR UNDRUGGABLE TARGETS

#### 3.1.1 TASK DEFINITION

We study fragment-level virtual screening on challenging protein targets. Given a protein pocket $p \in \mathcal{P}$—typically from an *undruggable* protein—and a fragment library $F = \{f_1, f_2, \ldots, f_N\}$, the goal is to identify a subset $F^+ \subseteq F$ of fragments that can form favorable non-covalent interactions with $p$. Each fragment $f_i$ is a chemically valid substructure derived from a drug-like molecule via synthetically accessible disconnections (e.g., BRICS rules). Compared to conventional ligands, fragments have lower molecular weight, surface area, and fewer functional groups, resulting in weak and localized binding. Yet this simplicity allows them to access shallow, flexible, or cryptic sites—precisely those found in undruggable pockets.

This setting presents unique challenges: binding signals are subtle, and standard screening methods optimized for full ligands often fail to prioritize fragments. Effective fragment retrieval therefore demands both dedicated benchmarks and tailored models.

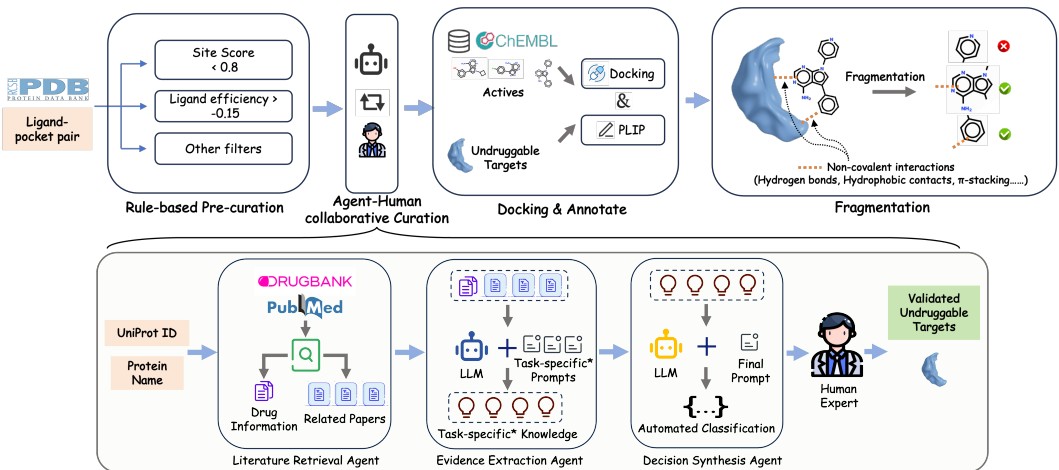

Figure 2: **Overview of the data curation pipeline for FragBench.** Starting from ligand–pocket pairs in the Protein Data Bank (PDB), we apply rule-based filters (e.g., site score, ligand efficiency) followed by a collaborative agent–human curation process to select high-quality protein–ligand complexes. For challenging targets from ChEMBL, docking and PLIP-based interaction analysis are used to annotate fragment-level contacts. Valid fragments are extracted based on interaction patterns (e.g., hydrogen bonding, hydrophobic contacts). To identify undruggable targets, we further use a literature-grounded reasoning pipeline that retrieves UniProt entries, mines PubMed/DrugBank evidence via LLM agents, and synthesizes task-specific knowledge. Human experts verify final decisions.

We construct a benchmark comprising undruggable targets, where each pocket $p$ is paired with a fragment set $F = F^+ \cup F^-$, containing known *binders* $F^+$ from experimental protein–ligand complexes and presumed *non-binders* $F^-$. Given a scoring model $s : F \to \mathbb{R}$ that ranks fragments for a fixed pocket $p$, we evaluate its ability to prioritize true binders using early recognition metrics: the **Enrichment Factor** (EF@$k$) and **BEDROC** (Truchon & Bayly, 2007).

EF@$k$ measures fold enrichment over random selection:

$$\text{EF@}k = \frac{\text{\# positives in top-}k}{\frac{k}{|F|} \cdot |F^+|}.$$

BEDROC emphasizes early retrieval in the full ranking:

$$\text{BEDROC}_\alpha = \frac{1 - e^{-\alpha}}{1 - e^{-\alpha R}} \sum_{i=1}^{|F^+|} e^{-\alpha \cdot \frac{r_i}{R}},$$

where $R = |F|$ is the total number of fragments and $r_i$ is the rank of the $i$-th positive.

### 3.1.2 RULE-BASED PRE-CURATION

Although an estimated 85% of the human proteome is considered *undruggable*, structural data in the Protein Data Bank (PDB) (Berman et al., 2000) is heavily skewed toward druggable proteins with well-formed pockets. Moreover, no existing database systematically catalogs undruggable targets. To bridge this gap, we developed a rule-based pipeline to extract challenging pocket–ligand pairs from PDB by combining structural heuristics with ligand efficiency filters.

We began with all protein–ligand complexes in PDB and excluded trivial cases involving covalent ligands, nucleic acid proximity (within 6Å), or small pockets (fewer than 10 residues), yielding 87,425 pairs. We then assessed each pocket using `SiteMap` (Halgren, 2009), retaining those with a site score below 0.8—indicative of small or poorly enclosed binding sites.

In parallel, we computed ligand efficiency (LE) as docking score per heavy atom:

$$\text{LE}(l) = \frac{S(l)}{\text{HA}(l)},$$

where $S(l)$ is the Glide docking score and $HA(l)$ is the heavy atom count. Pairs with $LE(l) > -0.15$ were prioritized, reflecting weak binding normalized by size.

The final candidate set is:

$$\mathcal{C} = \{(p, l) \in \mathcal{D}_{\text{PDB}} \mid \text{SiteScore}(p) < 0.8 \wedge \text{LE}(l) > -0.15\},$$

resulting in 1,387 structurally challenging pocket–ligand pairs for expert review.

### 3.1.3 MULTI-AGENT FRAMEWORK FOR TARGET CURATION

To construct an evidence-grounded benchmark of undruggable targets, we introduce a modular multi-agent framework composed of retrieval, extraction, synthesis, and expert validation components. Formally, given a protein target $t \in \mathcal{T}$ (where $\mathcal{T}$ is the set of UniProt-annotated human proteins), the system retrieves a corpus $\mathcal{D}_t$ of relevant documents by querying DrugBank (Knox et al., 2024) and PubMed (White, 2020). This retrieval is modeled as a mapping $\mathcal{R} : t \mapsto \mathcal{D}_t$, where $\mathcal{D}_t = \{d_1, d_2, \ldots, d_n\}$ includes abstracts and metadata related to $t$'s druggability, clinical development, and structural properties. Each document $d_i \in \mathcal{D}_t$ is then processed by an LLM-based extraction agent $\mathcal{E}$ with task-specific prompting, producing structured tuples $(e_i, c_i)$ where $e_i$ is an evidence type (e.g., "shallow pocket", "fragment hit", "undruggable domain" and $c_i$ is a citation. The agent enforces output consistency via schema-constrained decoding and validation heuristics. A synthesis agent $\mathcal{S}$ aggregates extracted tuples $\{(e_i, c_i)\}_{i=1}^n$ and metadata features $m_t$ from Drug-Bank to compute a provisional classification $\hat{y}_t \in \{\text{druggable}, \text{undruggable}, \text{FBDD-reported}\}$ with supporting citations. Conflicts or ambiguous cases (e.g., conflicting evidence of druggability and FBDD) are resolved via deterministic resolution rules or flagged for human audit. Finally, a domain expert validation step $\mathcal{V} : \hat{y}_t \mapsto y_t$ confirms or corrects each label $\hat{y}_t$, producing the final ground-truth annotation $y_t$ used in our benchmark. This human-in-the-loop step is essential for resolving nuanced biological edge cases, such as disordered proteins with low-confidence fragment data.

Overall, this framework achieves high-throughput and structured curation of undruggable targets, with an average of 218, 34, and 25 relevant PubMed documents per target for druggability, undruggability, and FBDD evidence respectively. The resulting benchmark provides structured evidence provenance and supports downstream model evaluation under realistic biological constraints.

### 3.1.4 FRAGMENT CONSTRUCTION

To support fragment-level learning, we constructed a dataset of fragment–pocket interactions for curated undruggable targets. Active ligands were retrieved from ChEMBL (Gaulton et al., 2012) with strict assay-based filtering described in G , and each molecule $m \in \mathcal{M}$ was decomposed into synthetically accessible fragments using the BRICS algorithm (Degen et al., 2008):

$$\mathcal{F}(m) = \{f_1, f_2, \ldots, f_k\}.$$

Fragments with $8 \leq HA(f_i) \leq 24$ were retained to match common fragment library constraints (e.g., Enamine REAL (Shivanyuk et al., 2007)). Redundancy was reduced via fingerprint-based clustering, details are in H.

Positive fragment labels were generated by redocking each ligand into its native pocket using Glide, followed by non-covalent interaction detection with PLIP (Salentin et al., 2015). To obtain *high-confidence* positive examples, we adopted a conservative consensus strategy: a fragment was labeled as positive only if it (i) formed at least two distinct non-covalent interactions with the pocket in a given docking pose, and (ii) this multi-interaction pattern was reproducibly observed across 3 independent docking replicates. Formally, let $\mathcal{A}(f)$ denote the set of atoms belonging to fragment $f$, and $\mathcal{I}(a, p)$ be the number of non-covalent interactions formed by atom $a$ with pocket $p$. A fragment is considered positive if $\left|\{a \in \mathcal{A}(f) \mid \mathcal{I}(a, p) > 0\}\right| \geq 2$ and this condition holds consistently across three docking replicates.

We quantitatively assessed the accuracy of this labeling strategy and examined how the number of docking replicates and interaction thresholds affect label reliability; details are provided in Appendix B.

Negatives were sampled randomly at a 1:90 positive-to-negative ratio from a fragment pool. The resulting **FragBench** dataset spans 54 targets, each associated with an average of 84.37 positive and

7593.33 negative fragments, providing the first standardized benchmark for fragment-level screening against challenging protein pockets. Comprehensive information on 54 targets, including their protein name, Uniprot ID and associated disease indications is presented in Appendix I.

In Figure 1(c), we report a statistical characterization of FragBench pockets, which display reduced size and fewer residues relative to DUD-E targets, indicative of their shallow and flattened topology. Dscores further underscore the intrinsic challenges these targets pose for rational drug design, highlighting the fundamental differences between FragBench and traditional benchmarks.

## 3.2 FRAGCLIP: A CONTRASTIVE LEARNING FRAMEWORK FOR FRAGMENT RETRIEVAL

### 3.2.1 MULTI-GRANULAR CONTRASTIVE ALIGNMENT

The core task of this work is *fragment retrieval*: given a protein pocket, the model must identify fragments likely to bind. Directly learning from fragment–pocket pairs is challenging due to the small size, weak binding affinity, and context-dependence of fragments. Such training would provide sparse and noisy supervision.

To address this, we design a multi-encoder framework that jointly models protein pockets, fragments, and their parent molecules. The protein encoder $f_p$ maps 3D pocket structures into a latent space. The fragment encoder $f_f$ captures fine-grained chemical substructures relevant to binding. The molecule encoder $f_m$ provides scaffold-level context, serving as a structural and chemical bridge to regularize fragment representations and stabilize training.

To align representations across these three molecular granularities, we employ a set of contrastive objectives: *(i)* pocket–molecule alignment ($\mathcal{L}_{\text{p-m}}$) preserves scaffold-level semantics, *(ii)* pocket–fragment alignment ($\mathcal{L}_{\text{p-f}}$) provides direct supervision for fragment–pocket compatibility, and *(iii)* molecule–fragment alignment ($\mathcal{L}_{\text{m-f}}$) enforces internal consistency between fragments and their source molecules.

Each loss takes the following form:

$$\mathcal{L}_{a-b} = -\frac{1}{N} \sum_{i=1}^{N} \log \frac{\exp(\text{sim}(f_a(a_i), f_b(b_i))/\tau)}{\sum_{j=1}^{N} \exp(\text{sim}(f_a(a_i), f_b(b_j))/\tau)}, \tag{1}$$

where $(a, b) \in \{(\text{p}, \text{m}), (\text{p}, \text{f}), (\text{m}, \text{f})\}$ and $\text{sim}(\cdot, \cdot)$ denotes cosine similarity. The total loss is:

$$\mathcal{L}_{\text{align}} = \mathcal{L}_{\text{p-m}} + \lambda_1 \mathcal{L}_{\text{p-f}} + \lambda_2 \mathcal{L}_{\text{m-f}}. \tag{2}$$

For implementation, we adopt the UniMol architecture (Zhou et al., 2023), a 3D molecular representation model with SE(3)-equivariant attention. We use UniMol's pocket encoder for $f_p$, and its molecular encoder for both $f_m$ and $f_f$, enabling unified geometric representations across all modalities.

### 3.2.2 FUSION MECHANISM FOR FRAGMENT IMPORTANCE MODELING

A fundamental challenge in fragment retrieval is that fragment-level signals are inherently noisy (Bon et al., 2022). Many fragments within a molecule contribute little to binding, while others may form spurious or context-dependent interactions. Relying on all fragments equally can therefore dilute the discriminative cues needed for accurate retrieval, making it difficult for the model to identify which substructures truly drive binding.

To address this, we introduce a fusion mechanism that performs joint selection and filtering of fragment information. Given a molecule embedding $f_m(m)$ and its associated fragment embeddings $\{f_f(f_i)\}_{i=1}^{k}$, a cross-attention module highlights fragments most relevant to binding while downweighting less informative ones. The attention output is concatenated with the molecule embedding and passed through a multilayer perceptron (MLP) to yield a fused representation:

$$z_{\text{fusion}} = \text{MLP}\Big(f_m(m) \, \| \, \text{Attn}(f_m(m), \{f_f(f_i)\}_{i=1}^{k})\Big). \tag{3}$$

This fused embedding is trained to align with the pocket representation via contrastive loss:

$$\mathcal{L}_{\text{fusion}} = -\frac{1}{N} \sum_{i=1}^{N} \log \frac{\exp(\text{sim}(f_p(p_i), z_{\text{fusion},i})/\tau)}{\sum_{j=1}^{N} \exp(\text{sim}(f_p(p_i), z_{\text{fusion},j})/\tau)}. \tag{4}$$

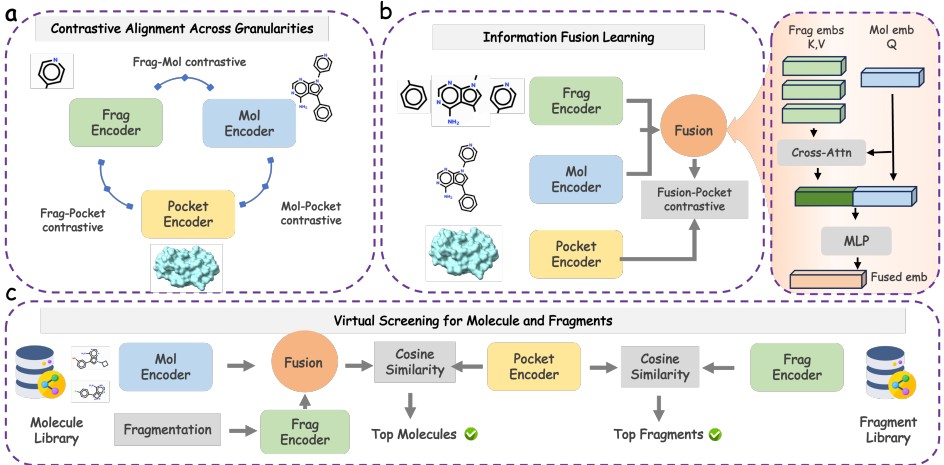

Figure 3: **Overview of the FRAGCLIP framework.** (a) *Contrastive Alignment Across Granularities*: three encoders model pockets, fragments, and molecules, aligned via multi-view contrastive losses to capture both fine-grained fragment–pocket interactions and scaffold-level semantics. (b) *Information Fusion Learning*: molecule and fragment embeddings are fused via cross-attention, enhancing fragment representation with contextual molecular information and enabling contrastive alignment with pockets. (c) *Fragment and Molecule Screening*: during inference, pockets are used to retrieve both top-scoring molecules and fragments via cosine similarity in the shared embedding space.

By emphasizing informative fragments and suppressing noise, the fusion module refines fragment embeddings into more discriminative signals for retrieval, highlighting binding-relevant substructures and strengthening fragment-level representation learning.

# 4 EXPERIMENTS

## 4.1 EXPERIMENT SETTINGS

**Tasks and Datasets**

Our primary task is *fragment retrieval* on **FragBench**, a benchmark of curated undruggable targets with shallow or cryptic pockets. In addition to FragBench, we also construct fragment-level retrieval benchmarks for several classical virtual-screening datasets, including **DUD-E**, **LIT-PCBA**, and **DEKOIS 2.0**. For each dataset, we repeat docking using Glide and label fragment–pocket interactions with PLIP, designating fragments that form at least two non-covalent interactions consistently as positives. Detailed dataset statistics and construction procedures are provided in Appendix E.

For model training we use **PDBbind**. To prevent information leakage from test targets, we remove from the training pool all protein–ligand complexes whose sequence identity to any target in Frag-Bench, DUD-E, LIT-PCBA, or DEKOIS 2.0 exceeds 90%. After filtering, the training and validation sets contain 14,223 and 744 protein–ligand pairs respectively. We further analyze the impact of different levels of sequence-homology filtering on fragment retrieval performance, and the complete results are provided in the Appendix.

**Evaluation Metrics** We report standard virtual screening metrics, focusing on early recognition performance. Specifically, we evaluate models using AUC, Enrichment Factor at top-$k$ (EF@$k$), and BEDROC. All results are averaged across targets.

**Baselines.** We compare **FragCLIP** with both classical docking/scoring methods and recent learning-based rescoring models. Glide-SP (Yang et al., 2021) and AutoDock Vina (Eberhardt et al., 2021) serve as standard docking baselines. We also include the widely-used machine learning scor-

ing function **RF-Score** (Ballester & Mitchell, 2010), and two high-performing structure-based scoring methods that explicitly model 3D geometry and residue–atom interactions, **RTMScore** (Shen et al., 2022) and **EquiScore** (Cao et al., 2023). Among learning-based baselines, **DrugCLIP** (Gao et al., 2023) and **LigUnity** (Feng et al., 2025) align pocket–ligand pairs via contrastive representation learning for retrieval and screening.

## 4.2 RESULTS ON FRAGBENCH (UNDRUGGABLE TARGETS)

We evaluate our method on the FragBench dataset, comparing against classical docking tools (Vina, Glide), machine learning models, and a recent contrastive learning baseline. As shown in Table 1, classical docking-based approaches completely fail in the fragment-level virtual screening setting, with Vina producing no meaningful ranking results and Glide achieving only marginal enrichment (EF@0.5% of 1.86 and BEDROC of 0.03). DrugCLIP improves performance slightly (EF@0.5% = 4.11). Our proposed method, **FragCLIP**, achieves the highest performance across all metrics, with an BEDROC of **0.12**, and EF@0.5% of **6.85**. These results underscore the importance of our framework: by explicitly modeling the fragment–pocket interaction in a contrastive and fragment-centric manner, FragCLIP substantially outperforms both docking and prior learning-based methods. This demonstrates the effectiveness of our design for fragment-level recognition.

Table 1: Performance comparison on FragBench. Results are averaged over all targets.

| Method | AUROC | BEDROC | EF@0.5% | EF@1% | EF@2% | EF@5% |
|---|---|---|---|---|---|---|
| Vina | 0.476 | 0.025 | 1.665 | 1.419 | 1.208 | 1.113 |
| Glide | $0.597_{0.009}$ | $0.034_{0.007}$ | $1.862_{0.543}$ | $1.825_{0.768}$ | $1.821_{0.422}$ | $1.712_{0.285}$ |
| RFscore | 0.457 | 0.025 | 1.665 | 1.419 | 1.469 | 1.113 |
| RTM Score[†] | 0.571 | 0.094 | 1.896 | 1.997 | 1.940 | 1.824 |
| EquiScore[†] | 0.581 | 0.105 | 4.039 | 3.331 | 2.638 | 2.049 |
| LigUnity[†] | 0.505 | 0.089 | 4.262 | 3.562 | 2.933 | 2.087 |
| DrugCLIP (90%) | $0.597_{0.027}$ | $0.080_{0.003}$ | $4.110_{0.056}$ | $3.203_{0.121}$ | $2.660_{0.072}$ | $2.067_{0.051}$ |
| FragCLIP (90%) | $\mathbf{0.593}_{0.018}$ | $\mathbf{0.115}_{0.003}$ | $\mathbf{6.853}_{0.582}$ | $\mathbf{5.797}_{0.258}$ | $\mathbf{4.510}_{0.163}$ | $\mathbf{3.000}_{0.161}$ |

*Subscripts denote standard deviations across three independent runs.
†Evaluated using the original checkpoint without homology filtering on the test set.

## 4.3 PERFORMANCE ON OTHER FRAGMENT BENCHMARKS

To rigorously evaluate FragCLIP across varying levels of difficulty, we conducted experiments on three **fragment-version** benchmarks: DUD-E, Dekois, and LIT-PCBA. For the construction of these datasets and detailed statistics, please refer to Appendix E. While DUD-E and Dekois represent standard benchmarks with well-characterized targets, LIT-PCBA poses a challenging dataset.

Remarkably, FragCLIP consistently achieves the best Enrichment Factor (EF) scores across all three datasets, demonstrating its superior capability in early recognition regardless of the benchmark difficulty. On the standard DUD-E and Dekois datasets, FragCLIP dominates with EF@0.5% scores of **20.317** and **17.963**, respectively, substantially outperforming baselines like LigUnity and DrugCLIP. Even on the challenging LIT-PCBA dataset, FragCLIP still secures the highest enrichment performance across all thresholds (e.g., **3.437** EF@0.5% vs. 2.939 for RTMScore). This consistent superiority in metrics highlights FragCLIP's robustness and practical value in prioritizing active fragments for virtual screening tasks.

## 4.4 FRAGMENT-AIDED MOLECULE RETRIEVAL VIA FUSION

We explored whether informative fragment-level signals could enhance **molecule-level retrieval**, results shown in Table 3. Our findings suggest that incorporating fragment supervision during training improves the quality of molecular representations. Specifically, by introducing fragment-level contrastive learning but performing retrieval solely using the molecule encoder, we observed an improvement at EF1% from **31.87** to **33.56**.

Table 2: Performance comparison on DUD-E, Dekois, and LIT-PCBA benchmarks (fragment version). Results are averaged over all targets.

| Method | AUROC | BEDROC | EF@0.5% | EF@1% | EF@2% | EF@5% |
|---|---|---|---|---|---|---|
| **DUD-E (fragment version)** | | | | | | |
| Vina | 0.521 | 0.062 | 4.805 | 3.897 | 3.155 | 2.323 |
| Glide | $0.621_{0.009}$ | $0.087_{0.010}$ | $7.535_{1.458}$ | $5.795_{1.014}$ | $4.153_{0.577}$ | $2.807_{0.322}$ |
| RTMScore[†] | 0.454 | 0.007 | 1.818 | 1.607 | 1.448 | 1.353 |
| EquiScore[†] | 0.658 | 0.137 | 4.442 | 3.569 | 3.217 | 2.726 |
| LigUnity[†] | $0.616_{0.129}$ | $0.194_{0.136}$ | 19.493 | 14.049 | 9.078 | 4.891 |
| DrugCLIP (90%) | $0.642_{0.019}$ | $0.136_{0.003}$ | $12.013_{0.326}$ | $9.333_{0.220}$ | $6.843_{0.137}$ | $4.320_{0.140}$ |
| FragCLIP (90%) | $\mathbf{0.761}_{0.015}$ | $\mathbf{0.227}_{0.007}$ | $\mathbf{20.317}_{1.020}$ | $\mathbf{16.012}_{0.702}$ | $\mathbf{11.307}_{0.293}$ | $\mathbf{6.883}_{0.136}$ |
| **Dekois (fragment version)** | | | | | | |
| Vina | 0.546 | 0.053 | 3.619 | 3.022 | 2.770 | 2.330 |
| Glide | $0.630_{0.010}$ | $0.065_{0.012}$ | $4.694_{1.964}$ | $4.098_{1.151}$ | $3.297_{0.736}$ | $2.597_{0.369}$ |
| RFscore | 0.530 | 0.038 | 2.790 | 2.276 | 1.952 | 1.544 |
| RTMScore[†] | 0.506 | 0.069 | 2.153 | 1.819 | 1.569 | 1.274 |
| EquiScore[†] | 0.658 | 0.138 | 3.904 | 3.718 | 3.376 | 2.706 |
| DrugCLIP (90%) | $0.640_{0.019}$ | $0.113_{0.002}$ | $8.607_{0.170}$ | $7.523_{0.145}$ | $5.937_{0.080}$ | $4.027_{0.175}$ |
| FragCLIP (90%) | $\mathbf{0.750}_{0.013}$ | $\mathbf{0.213}_{0.004}$ | $\mathbf{17.963}_{0.764}$ | $\mathbf{14.710}_{0.447}$ | $\mathbf{10.907}_{0.157}$ | $\mathbf{6.773}_{0.163}$ |
| **LIT-PCBA (fragment version)** | | | | | | |
| Vina | 0.492 | 0.025 | 1.483 | 1.250 | 1.261 | 1.324 |
| Glide | 0.546 | 0.018 | 0.528 | 0.939 | 1.038 | 0.927 |
| RFscore | 0.456 | 0.020 | 1.366 | 1.269 | 0.876 | 0.865 |
| RTMScore[†] | 0.567 | **0.095** | 2.939 | 2.818 | 2.111 | 1.698 |
| EquiScore[†] | **0.597** | 0.086 | 2.446 | 1.981 | 1.643 | 1.470 |
| DrugCLIP (90%) | $0.560_{0.023}$ | $0.032_{0.003}$ | $1.823_{0.344}$ | $1.550_{0.164}$ | $1.817_{0.219}$ | $1.673_{0.225}$ |
| FragCLIP (90%) | $0.575_{0.029}$ | $0.050_{0.004}$ | $\mathbf{3.437}_{0.125}$ | $\mathbf{2.857}_{0.304}$ | $\mathbf{2.517}_{0.324}$ | $\mathbf{2.280}_{0.226}$ |

*Subscripts denote standard deviations across three independent runs.
†Evaluated using the original checkpoint/settings without homology filtering on the test set where applicable.

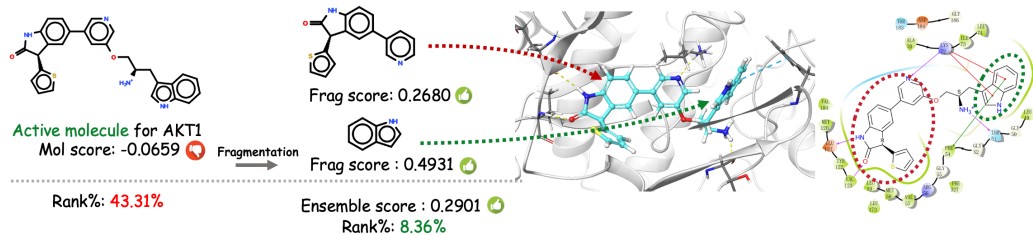

Figure 4: Case of fragment-level interactions boost molecule ranking. Two informative fragments with strong pocket interactions (FragScores: 0.2680, 0.4931) were identified from an active AKT1 ligand poorly scored by the molecule encoder (MolScore: –0.0659, Rank%: 43.31%). Incorporating these fragments via ensemble raised the final score (Ensemble score: 0.2901) and improved the ranking to 8.36%, illustrating the value of fragment-level signals in enhancing molecule retrieval.

Building on this, we further investigated how fragment information could be integrated at inference time. We combined fragment-level scores with molecule-level scores through both fusion and ensembling strategies to obtain a more fine-grained assessment. In particular, we implemented a multi-granularity ensemble where the final score is computed as:

$$\text{Score} = \text{MolScore} + \alpha \cdot \text{FragScore} + \beta \cdot \text{FusionScore}$$

with hyperparameters $\alpha = \beta = 0.8$. This approach yielded a gain in performance, achieving an EF1% of 37.23—demonstrating the utility of learned fragment representations in complementing

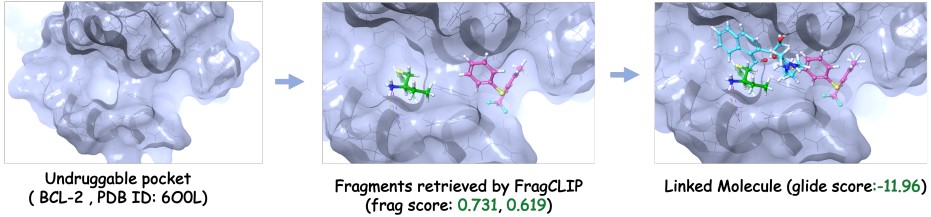

Figure 5: Fragment linking for BCL-2 using FragCLIP and DiffLinker. Left: The undruggable binding pocket of BCL-2, characterized by a smooth and shallow surface. Middle: Two high-scoring fragments retrieved by FragCLIP, each occupying distinct subpockets and serving as anchors. Right: A linked molecule generated by DiffLinker, maintaining favorable interactions across the pocket and achieving a strong predicted binding affinity.

molecular signals. Figure 4 highlights a representative case where a molecule initially ranked poorly by the molecule encoder was significantly re-ranked due to strong fragment-level evidence.

Table 3: Molecule-level virtual screening performance on the DUD-E dataset.

| Method | AUC ↑ | BEDROC ↑ | EF ↑ | | |
|---|---|---|---|---|---|
| | | | 0.5% | 1% | 5% |
| Glide-SP | 76.70 | 40.70 | 19.39 | 16.18 | 7.23 |
| Vina | 71.60 | – | 9.13 | 7.32 | 4.44 |
| NN-score | 68.30 | 12.20 | 4.16 | 4.02 | 3.12 |
| RFscore | 65.21 | 12.41 | 4.90 | 4.52 | 2.98 |
| Pafnucy | 63.11 | 16.50 | 4.24 | 3.86 | 3.76 |
| OnionNet | 59.71 | 8.62 | 2.84 | 2.84 | 2.20 |
| Planet | 71.60 | – | 10.23 | 8.83 | 5.40 |
| DrugCLIP | 80.93 | 50.52 | 38.07 | 31.89 | 10.66 |
| FragCLIP (w/o Fusion) | 84.76 | 53.61 | 40.64 | 33.56 | 11.39 |
| **FragCLIP** | **85.44** | **59.32** | **42.93** | **37.23** | **12.45** |

## 4.5 FRAGMENT LINKING ON BCL-2

Fragments from virtual screening typically require growing or linking to form drug-like molecules. To demonstrate practical utility, we have performed fragment linking on the undruggable target BCL-2. FragCLIP retrieved 30 candidate fragments, which were docked with Glide to identify anchor conformations. Using DiffLinker, we generated complete molecules by linking fragment pairs. As shown in Fig. 5, two high-scoring fragments occupied distinct subpockets and served as effective anchors, yielding a linked molecule with a Glide score of **-11.96**. This case highlights the potential of combining FragCLIP with generative models for designing novel compounds against challenging targets.

## 5 CONCLUSION

We present FRAGBENCH, the first benchmark for fragment-level retrieval on *undruggable* protein targets. Our findings reveal that docking-based methods struggle with fragments, especially in shallow or cryptic pockets. To address this, we introduce FRAGCLIP, a cross-modal framework that learns aligned representations via multi-view contrastive pretraining. By integrating global context and local interactions, FRAGCLIP outperforms existing baselines across all metrics. Together, FRAGBENCH and FRAGCLIP establish a foundation for fragment-centric modeling in FBDD, advancing drug discovery for targets beyond the reach of conventional methods.

ACKNOWLEDGMENTS

This work is supported by Innovative Drug Research and Development-National Science and Technology Major Project (No.2025ZD1802501) and Beijing Frontier Research Center for Biological Structure Fundings.

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

## A LLM USAGE STATEMENT:

GPT-4o was used exclusively for language polishing, including grammar correction and clarity enhancement. All scientific content and analyses were conducted independently of LLMs.

## B EVALUATION OF FRAGMENT–POCKET LABEL QUALITY AND THRESHOLD SELECTION

To quantify the reliability of docking-based fragment interaction labelling and to determine appropriate interaction and consensus thresholds for defining positive fragments, we conducted a evaluation on 500 PDBbind complexes. For each complex, fragment labels derived from the experimental co-crystal pose using PLIP were treated as the reference, while labels generated from three independent random seeds Glide docking replicates served as predictions.

**Experimental design.** Two factors were varied to study their impact on label accuracy:

- **Interaction threshold:** a fragment is considered positive in a given docking run if it forms at least $k \in \{1, 2, 3\}$ distinct non-covalent interactions with the pocket.
- **Replicate consensus:** a fragment is considered positive overall if it is predicted positive in at least $m \in \{1, 2, 3\}$ of the docking runs.

This yields nine labeling configurations $(k, m)$, covering both lenient (low $k$ or $m$) and conservative (high $k$ and $m$) regimes. Precision, recall, and F1 score were computed by comparing docking-derived labels against the reference crystal-derived labels on a fragment-by-fragment basis. Heatmaps and trend plots summarizing the results are shown in Fig. 6.

**Quantitative results.** Table 4 summarizes the performance across all configurations. Overall, the results exhibit a clear and monotonic precision–recall trade-off. Increasing the replicate-consensus requirement consistently improves precision, while higher interaction thresholds further tighten the definition of a positive fragment. As expected, stricter criteria reduce recall, but they substantially suppress false positives. In the context of constructing positive labels for downstream learning—where precision is the primary concern—configurations enforcing both multiple interactions and cross-replicate consistency provide the most reliable supervision.

**Interpretation.** These analyses highlight that:

1. Docking-derived fragment labels align closely with crystal-structure interaction patterns across all threshold settings, demonstrating that docking provides a robust and trustworthy source of weak supervision in this context.
2. Multi-replicate agreement effectively filters out seed-dependent fluctuations and stabilizes interaction assignments.
3. Requiring multiple interactions preferentially selects structurally meaningful and well-supported contact motifs, improving label reliability even at the cost of lower recall.

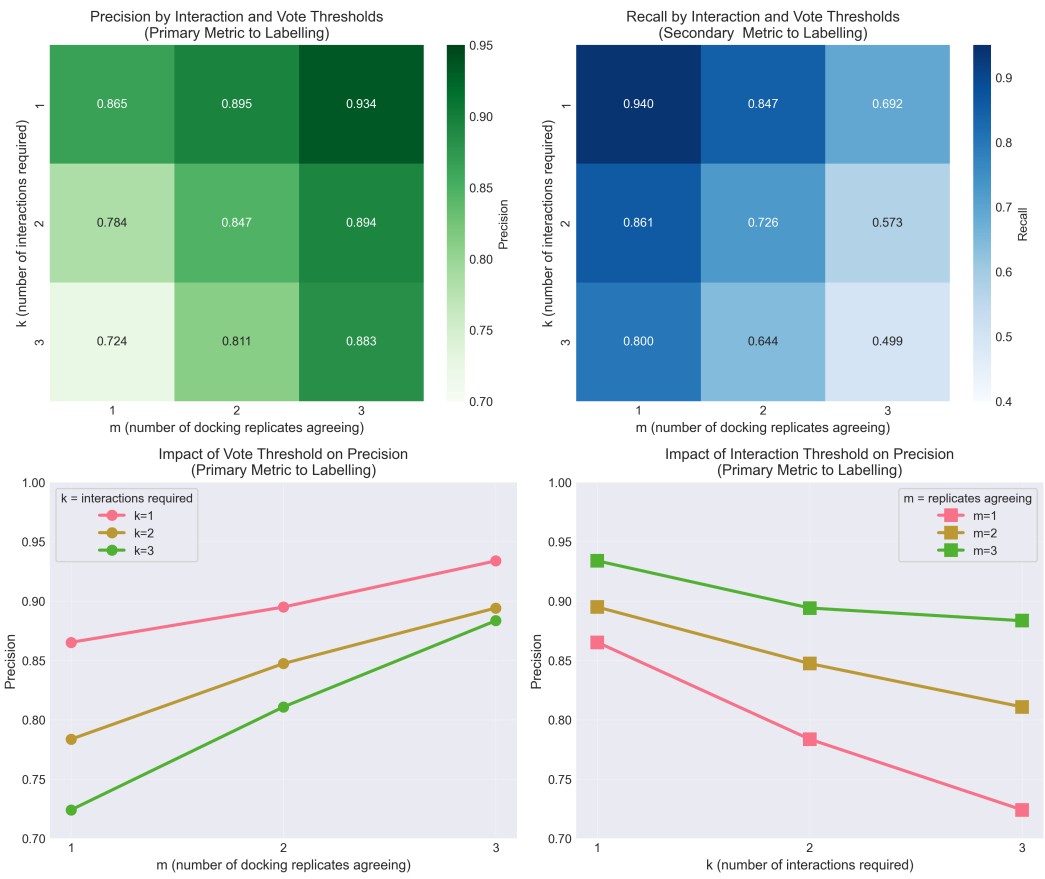

Figure 6: Precision and recall of fragment labels under different interaction and voting thresholds.

Table 4: Label accuracy across interaction and replicate-consensus thresholds, comparing docking-derived labels to crystal-based labels.

| Setting | Precision | Recall | F1 |
|---------|-----------|--------|-----|
| $k=1, m=3$ | 0.934 | 0.692 | 0.795 |
| $k=1, m=2$ | 0.895 | 0.847 | 0.870 |
| $k=1, m=1$ | 0.865 | 0.940 | 0.901 |
| $k=2, m=3$ | 0.894 | 0.573 | 0.698 |
| $k=2, m=2$ | 0.847 | 0.726 | 0.782 |
| $k=2, m=1$ | 0.784 | 0.861 | 0.821 |
| $k=3, m=3$ | 0.883 | 0.499 | 0.638 |
| $k=3, m=2$ | 0.811 | 0.644 | 0.718 |
| $k=3, m=1$ | 0.724 | 0.800 | 0.760 |

Based on these observations, we adopt the setting $k=2$ and $m=3$ for constructing the benchmark. Under this configuration, the labeled positive fragments achieve a precision of 89.4%, which, in the absence of large-scale experimental fragment–binding datasets, provides sufficiently reliable supervision for building a high-quality benchmark.

## C ABLATION STUDIES

To assess the contribution of each architectural component in FragCLIP, we perform ablations shown in Table 5 . We evaluate two major variants:

**No Fusion Module.** In this setting, we retain all three encoders—pocket, fragment, and molecule—but remove the fusion module responsible for cross-modal interaction between the fragment and molecule representations. This modification isolates the effect of the fusion mechanism.

**No Molecule Modality.** Here we disable the molecule encoder entirely and perform contrastive learning only between pocket and fragment representations. Architecturally, this reduces FragCLIP to a two-encoder framework analogous to DrugCLIP, but trained specifically on fragment-level data. Performance decreases notably compared to the full model, indicating that molecular context provides valuable structural and chemical cues that improve fragment discrimination beyond pocket–fragment geometry alone.

Overall, the ablation results clearly show that both the fusion module and the molecule modality play essential and complementary roles. The fusion module enables richer cross-modal reasoning, while the molecule encoder provides contextual constraints that guide fragment-level preferences. Removing either component consistently degrades performance, confirming that the full tri-modal design of FragCLIP is proper for fragment retrieval.

Table 5: Ablation study on model architecture.

| Setting | AUROC | BEDROC | EF@0.5% | EF@1% | EF@2% |
|---|---|---|---|---|---|
| FragCLIP | $0.593_{0.018}$ | $0.115_{0.003}$ | $6.853_{0.582}$ | $5.797_{0.258}$ | $4.510_{0.163}$ |
| No Fusion Module | $0.585_{0.006}$ | $0.105_{0.007}$ | $6.428_{0.241}$ | $5.072_{0.394}$ | $4.174_{0.271}$ |
| No Molecule Modality | $0.584_{0.002}$ | $0.106_{0.006}$ | $6.170_{0.429}$ | $5.296_{0.217}$ | $4.049_{0.274}$ |

*Subscripts denote standard deviations across three independent runs.

## D    IMPACT OF SEQUENCE-HOMOLOGY FILTERING ON FRAGMENT RETRIEVAL

To evaluate how sequence similarity between training and test proteins affects fragment retrieval performance, we construct three increasingly stringent de-homogenized training sets by filtering PDBbind according to protein sequence identity with respect to all test targets in test dateset. We consider three levels of sequence-homology filtering when constructing the training set:

**PDB-ID–deduplicated setting (Default).** Only protein–ligand complexes with the exact same PDB ID as any test target are removed. This corresponds to the 100% identity setting and serves as our full training set.

**90% / 60% / 30% sequence-identity–filtered setting.** All training proteins whose sequence identity to any test protein exceeds 90% / 60% / 30% are removed.

Table 6 reports the resulting training and validation sizes under each setting.

Table 6: Training/validation set sizes under different sequence-identity thresholds.

| Identity Setting | Train Pairs | Val Pairs | Change w.r.t. Full Set |
|---|---|---|---|
| 100% (Full) | 17,315 | 919 | baseline |
| 90% Filtered | 14,223 | 744 | −3,267 (17.92%) |
| 60% Filtered | 12,487 | 657 | −5,090 (27.91%) |
| 30% Filtered | 10,376 | 556 | −7,302 (40.05%) |

Overall, stricter sequence-homology filtering leads to a clear decline in fragment retrieval performance (Table 7). The largest drop occurs when moving from the full PDB-ID–deduplicated setting (100%) to the 90% identity threshold, indicating that removing highly homologous proteins has the strongest impact. Further tightening the threshold from 90% to 60% produces a smaller relative decrease. Notably, when the threshold is further restricted to 30%, the magnitude of the performance drop becomes even more marginal, confirming that the sensitivity to homology reduction diminishes at stricter levels.

Taken together, these results show that FragCLIP is sensitive to homology filtering—as expected. The 90% sequence-identity setting provides a practical balance between avoiding information leakage and maintaining sufficient training diversity, and is therefore adopted as our primary de-homogenized evaluation condition in the main paper.

Table 7: Fragment retrieval performance of FragCLIP under different sequence-homology filtering thresholds on FragBench.

| Setting | AUROC | BEDROC | EF@0.5% | EF@1% | EF@2% | EF@5% |
|---|---|---|---|---|---|---|
| FragCLIP (100%) | $0.637_{0.007}$ | $0.147_{0.006}$ | $10.741_{0.324}$ | $8.155_{0.292}$ | $5.737_{0.137}$ | $3.767_{0.077}$ |
| FragCLIP (90%) | $0.593_{0.018}$ | $0.115_{0.003}$ | $6.853_{0.582}$ | $5.797_{0.258}$ | $4.510_{0.163}$ | $3.000_{0.161}$ |
| FragCLIP (60%) | $0.593_{0.022}$ | $0.091_{0.010}$ | $4.843_{0.372}$ | $4.283_{0.750}$ | $3.569_{0.449}$ | $2.660_{0.301}$ |
| FragCLIP (30%) | $0.554_{0.008}$ | $0.079_{0.006}$ | $4.337_{0.186}$ | $3.977_{0.221}$ | $3.213_{0.255}$ | $2.270_{0.053}$ |

*Subscripts denote standard deviations across three independent runs.

## E    CONSTRUCTION OF FRAGMENT-LEVEL BENCHMARKS FOR DUD-E, DEKOIS 2.0, AND LIT-PCBA

In addition to FragBench, we construct fragment-level retrieval benchmarks for three widely used structure-based virtual screening datasets: **DUD-E**, **DEKOIS 2.0**, and **LIT-PCBA**. Our goal is to provide fragment-based counterparts to these classical molecule-level benchmarks, enabling consistent evaluation of fragment retrieval methods across diverse protein families and pocket types.

**Fragment generation and interaction labeling.**    For each dataset, we begin from the original active and decoy molecules associated with each target. Each molecule is decomposed into synthetically accessible fragments using the BRICS algorithm. For every target, the protein pocket is extracted from its reference binding structure, and each fragment is docked into the pocket using Glide. Fragment–pocket interactions are then quantified using PLIP. A fragment is labeled as *positive* if it forms at least two distinct non-covalent interactions with the pocket in a docking pose and this interaction pattern is consistently reproduced across three independent docking replicates. In all three datasets, we applied the same fingerprint-based fragment clustering procedure used in FragBench to reduce redundancy and ensure diversity among the fragment candidates. To maintain consistency with the imbalance characteristics of the original molecule-level datasets (e.g., DUD-E's heavy active–decoy skew), we sample negatives at a fixed ratio of **1:90** relative to positives for each target.

**Dataset statistics.**    Table 8 reports the number of targets, positive fragments, negative fragments, and dataset-level averages for all three fragment benchmarks.

Table 8: Fragment-level dataset statistics for DUD-E, DEKOIS 2.0, and LIT-PCBA.

| Dataset | Targets | Positives | Negatives | Avg Pos/Target | Avg Neg/Target |
|---|---|---|---|---|---|
| DUD-E | 96 | 8,740 | 786,600 | 91.0 | 8,193.8 |
| DEKOIS 2.0 | 81 | 6,398 | 575,820 | 79.0 | 7,108.9 |
| LIT-PCBA | 15 | 1,256 | 113,040 | 83.7 | 7,536.0 |
| **Total** | 246 | 20,950 | 1,885,500 | 85.2 | 7,664.6 |

These fragment-level benchmarks complement FragBench and enable comprehensive evaluation of fragment retrieval performance across both druggable and challenging targets.

## F    IMPLEMENTATION DETAILS

All models were trained on 4 NVIDIA A100 GPUs (80GB) using a batch size of 20. We employed a linear learning rate warm-up schedule for the first 5% of training steps, followed by cosine decay. The protein and molecular encoders are implemented as SE(3)-equivariant 3D convolutional

neural networks following the Uni-Mol architecture (Zhou et al., 2023). All contrastive losses use a temperature parameter $\tau = 1$, and we optimize using the AdamW optimizer with weight decay of $1 \times 10^{-3}$. Models are trained for 100 epochs.

## G  CHEMBL DATA FILTERING PROTOCOL

To ensure the reliability and consistency of bioactivity data used in our benchmark, we applied a set of rigorous filtering criteria to extract high-confidence ligand–target interactions from ChEMBL. The selection process involved both assay-level and activity-level filters, focusing on well-annotated functional assays with validated outcomes. Specifically, the following conditions were enforced:

- **Assay Confidence and Type:** Only entries with `assays.confidence_score = 9`, indicating direct target assignment, and `assays.assay_type` being either 'B' (binding) or 'F' (functional) were retained.
- **Activity Validity:** We retained entries where `activities.data_validity_comment` was either `NULL` or 'Manually validated', ensuring manual curation.
- **Quantitative Bioactivity Measurements:** We included entries with either a valid combination of `standard_relation`, `standard_value`, and `standard_units`—where `standard_units` $\in \{\text{pM}, \text{nM}, \mu\text{M}\}$ and `standard_value` $\in [0.001, 1{,}000{,}000]$—or a textual `activity_comment` indicative of biological activity. The accepted comments include: 'active', 'weak activity', 'slightly active', 'slight inhibition', 'potent inhibitor', 'partially active', 'partial antagonist', 'partial agonist', 'non-competitive antagonist', 'no significant effect', 'no significant activity', 'no effect', 'no activity', 'no action', 'inverse agonist', 'irreversible antagonist', 'inhibition not detected', 'inactive', 'dose-dependent effect', 'antagonist', 'agonist', 'activator', as well as comments beginning with 'not active' or 'no inhibit'.
- **Standard Types:** Only the following potency-related measurement types were included:

$$\text{Standard\_Type} \in \{\text{IC}_{50}, \text{XC}_{50}, \text{EC}_{50}, \text{AC}_{50}, K_i, K_d, \text{ED}_{50}\}$$

- **Relation Operators:** To ensure comparability, we kept entries where:

$$\text{Standard\_Relation} \in \{=, <, >, \leq, \geq, \ll, \gg\}$$

## H  FRAGMENT CLUSTERING PROTOCOL

To ensure chemical diversity and reduce redundancy in the fragment pool, we performed clustering based on molecular similarity. Fragments were first filtered to ensure chemical validity—only molecules with valid SMILES, successful sanitization, and an atom count between 6 and 24 were retained.

For each valid fragment, we computed **feature-based circular fingerprints (FCFP6)** using RDKit with a radius of 3 and 4096 bits. Pairwise Tanimoto similarities were used to construct a distance matrix, followed by **hierarchical agglomerative clustering** with Ward linkage. A distance threshold of 0.9 was applied to define cluster membership.

From each resulting cluster, representative fragments were sampled to maintain diversity across the dataset. This clustering process was applied in parallel across all fragment sets using Python's `multiprocessing` utilities for scalability.

## I  FRAGBENCH TARGETS DETAILS

Table 9 presents the detailed information for the targets included in the FragBench dataset. For each entry, we provide the UniProt ID, protein name, functional classification, and the reference PDB structure. Structural properties include the SiteScore and AA Num, which denotes the number of

amino acid residues located within the 6Å binding pocket. Additionally, the table lists associated diseases to highlight the clinical relevance of each target.

Table 9: Targets and associated diseases formatted in dual-row style

| Uniprot | Protein Name | Class | SiteScore | AA Num | PDB |
|---------|--------------|-------|-----------|--------|-----|
| P00749 | Urokinase-type plasminogen activator | Enzyme | 0.799 | 23 | 1gi8 |
| **Diseases:** breast cancer, colorectal cancer, lung cancer, pancreatic cancer, rheumatoid arthritis, pulmonary fibrosis, neurodegenerative diseases | | | | | |
| P62993 | Growth factor receptor-bound protein 2 | Signaling / regulatory | 0.557 | 15 | 1x0n |
| **Diseases:** cervical cancer, colorectal cancer, chronic myeloid leukemia, hepatocellular carcinoma, prostate cancer (anti-androgen resistance) | | | | | |
| P00441 | Superoxide dismutase [Cu-Zn] | Enzyme | 0.372 | 10 | 2wz6 |
| **Diseases:** amyotrophic lateral sclerosis (ALS), Parkinson's disease, cardiovascular diseases, COVID-19, ischemic stroke | | | | | |
| P17931 | Galectin-3 | Other | 0.463 | 21 | 4bm8 |
| **Diseases:** NASH-related hepatic fibrosis, idiopathic pulmonary fibrosis, pancreatic ductal adenocarcinoma, IgA nephropathy, insulin resistance | | | | | |
| P14735 | Insulin-degrading enzyme | Enzyme | 0.783 | 19 | 4gs8 |
| **Diseases:** Type 2 diabetes mellitus, Alzheimer's disease, Cancer (various types), Neurodegenerative disorders involving amyloid-$\beta$ aggregation, Nonalcoholic fatty liver disease | | | | | |
| P19491 | Glutamate receptor 2 | Receptor | 0.428 | 18 | 4u23 |
| **Diseases:** Nicotine addiction, Alcohol use disorder, Schizophrenia and related psychoses, Alzheimers disease, Amyotrophic lateral sclerosis | | | | | |
| Q13822 | Autotaxin | Enzyme | 0.790 | 14 | 4zg9 |
| **Diseases:** idiopathic pulmonary fibrosis (IPF), systemic sclerosis (SSc), cancer (e.g., breast cancer, lung adenocarcinoma), cholestatic pruritus, cardiovascular and metabolic diseases | | | | | |
| P47929 | Galectin-7 | Other | 0.557 | 15 | 5h9q |
| **Diseases:** cancer (multiple types), SJS/TEN, preeclampsia, psoriasis, asthma | | | | | |
| Q9UIF8 | Bromodomain adjacent to zinc finger domain protein 2B | Transcription / epigenetic | 0.787 | 16 | 5or9 |
| **Diseases:** orthodontic-related gingival overgrowth | | | | | |
| P22734 | Catechol O-methyltransferase | Enzyme | 0.659 | 11 | 5pa0 |
| **Diseases:** breast cancer, systemic lupus erythematosus (SLE), autoimmune diseases involving Tfh dysregulation, allergic rhinitis, narcolepsy type 1 | | | | | |
| Q14145 | Kelch-like ECH-associated protein 1 | Signaling / regulatory | 0.762 | 12 | 5wiy |
| **Diseases:** non-small cell lung cancer, KRAS-driven lung adenocarcinoma, neurodegenerative diseases, nonalcoholic fatty liver disease, sepsis-associated ferroptosis | | | | | |
| P41182 | B-cell lymphoma 6 protein | Transcription / epigenetic | 0.547 | 12 | 6c3l |
| **Diseases:** breast cancer, autoimmune diseases, systemic lupus erythematosus, narcolepsy type 1, allergic rhinitis | | | | | |

*Continued on next page...*

| Uniprot | Protein Name | Class | SiteScore | AA Num | PDB |
|---|---|---|---|---|---|
| Q9NUW8 | Tyrosyl-DNA phosphodiesterase 1 | Enzyme | 0.792 | 20 | 6w4r |
| **Diseases:** non-small cell lung cancer, colorectal cancer, pancreatic ductal adenocarcinoma, lung adenocarcinoma, low-grade epilepsy-associated developmental tumors | | | | | |
| P01116 | GTPase KRas | Signaling / regulatory | 0.758 | 20 | 7acf |
| **Diseases:** Non-small cell lung cancer, colorectal cancer, pancreatic ductal adenocarcinoma, lung adenocarcinoma, adenomatoid odontogenic tumor | | | | | |
| O15151 | Protein Mdm4 | Signaling / regulatory | 0.726 | 20 | 7c3y |
| **Diseases:** hepatocellular carcinoma, renal cell carcinoma, colon cancer, breast cancer (including metastases), pulmonary fibrosis | | | | | |
| Q92793 | CREB-binding protein | Transcription / epigenetic | 0.351 | 16 | 8fup |
| **Diseases:** colorectal cancer, pancreatic ductal adenocarcinoma, chronic myeloid leukemia, neurodegenerative disorders, inflammatory diseases | | | | | |
| P16581 | E-selectin | Receptor | 0.481 | 20 | 8r5m |
| **Diseases:** acute myeloid leukemia, prostate cancer, pancreatic cancer, atherosclerosis, venous thrombosis | | | | | |
| P02879 | Ricin | Other | 0.750 | 13 | 8t9v |
| **Diseases:** ricin intoxication/poisoning | | | | | |
| P02769 | Albumin | Transport / carrier | 0.455 | 13 | 8wdd |
| **Diseases:** pancreatic carcinoma, gastric cancer with peritoneal metastasis, liver failure with systemic inflammation, inflammatory bowel disease, acute ischemic stroke | | | | | |
| P02766 | Transthyretin | Transport / carrier | 0.497 | 14 | 1eta |
| **Diseases:** Transthyretin amyloid cardiomyopathy , Hereditary transthyretin amyloidosis with polyneuropathy, Hereditary/mixed-phenotype ATTR amyloidosis, Wild-type (senile systemic) transthyretin amyloidosis | | | | | |
| P60568 | Interleukin-2 | Other | 0.684 | 17 | 1py2 |
| **Diseases:** multiple sclerosis, transplant rejection, high-risk neuroblastoma, metastatic melanoma, autoimmune hepatitis, autoimmune diseases (general), lymphoma, advanced/metastatic renal cell carcinoma | | | | | |
| P08254 | Stromelysin-1 | Enzyme | 0.752 | 16 | 1usn |
| **Diseases:** cancer (general, including mammary tumor models), metastatic melanoma, lung cancer, gastric cancer and gastric carcinogenesis associated with Helicobacter pylori infection | | | | | |
| Q92731 | Estrogen receptor beta | Receptor | 0.789 | 14 | 2fsz |
| **Diseases:** breast cancer, triple-negative breast cancer, prostate cancer, lung cancer, glioblastoma, colorectal cancer | | | | | |
| P56524 | Histone deacetylase 4 | Enzyme | 0.799 | 17 | 2vqv |
| **Diseases:** pulmonary arterial hypertension, myocardial infarction, cardiac fibrosis and cardiovascular disease, ischemic stroke, diabetic nephropathy | | | | | |
| P10275 | Androgen receptor | Receptor | 0.657 | 16 | 2ylo |
| **Diseases:** prostate cancer, metastatic castration-resistant prostate cancer, triple-negative breast cancer, androgenetic alopecia, hirsutism | | | | | |
| P42574 | Caspase-3 | Enzyme | 0.629 | 13 | 3dek |
| **Diseases:** Spinal cord injury, Mechanical-ventilation–induced diaphragm atrophy, Cancer (various types), Ischemia/reperfusion injury, Age-related macular degeneration | | | | | |
| Q00987 | E3 ubiquitin-protein ligase Mdm2 | Enzyme | 0.789 | 25 | 3lbk |
| **Diseases:** osteosarcoma, chronic myeloid leukemia, glioblastoma, melanoma, breast cancer | | | | | |
| P08235 | Mineralocorticoid receptor | Receptor | 0.690 | 15 | 3vhv |

| Uniprot | Protein Name | Class | SiteScore | AA Num | PDB |
|---|---|---|---|---|---|
| **Diseases:** hypertension (including resistant hypertension), heart failure, diabetic kidney disease, chronic kidney disease, primary aldosteronism | | | | | |
| P00747 | Plasminogen | Coagulation factor | 0.628 | 13 | 4cik |
| **Diseases:** fibrotic renal disease, thrombotic/fibrinolytic disorders (VTE, pulmonary embolism, myocardial infarction, ischemic stroke), cancers, Alzheimers disease, submacular hemorrhage | | | | | |
| P14740 | Dipeptidyl peptidase 4 | Enzyme | 0.786 | 26 | 4ffw |
| **Diseases:** type 2 diabetes mellitus, obesity, autoimmune diabetes, bullous pemphigoid, colorectal cancer | | | | | |
| P06802 | Ectonucleotide pyrophosphatase/PDE 1 | Enzyme | 0.794 | 25 | 4gtz |
| **Diseases:** calcium pyrophosphate deposition disease (CPPD), chondrocalcinosis, breast cancer stem cell generation | | | | | |
| P45452 | Collagenase 3 | Enzyme | 0.662 | 10 | 4l19 |
| **Diseases:** osteoarthritis, cancer (including breast and esophageal cancer), intestinal fibrosis in Crohns disease, pulmonary fibrosis, keratoconus | | | | | |
| Q8NB16 | Mixed lineage kinase domain-like protein | Signaling / regulatory | 0.726 | 14 | 4mwi |
| **Diseases:** dementia_in_Alzheimers_disease, non_alcoholic_fatty_liver_disease, amyotrophic_lateral_sclerosis, liver_fibrosis, diabetic_kidney_disease | | | | | |
| P03951 | Coagulation factor XI | Coagulation factor | 0.763 | 29 | 4na8 |
| **Diseases:** deep vein thrombosis, ischemic stroke, myocardial infarction, venous thromboembolism, cardioembolic stroke | | | | | |
| Q9BY41 | Histone deacetylase 8 | Enzyme | 0.748 | 16 | 4rn2 |
| **Diseases:** breast cancer, glioma, endometriosis, peritoneal fibrosis, Schistosoma mansoni infection (schistosomiasis) | | | | | |
| P08684 | Cytochrome P450 3A4 | Enzyme | 0.555 | 11 | 5a1p |
| **Diseases:** Affecting drug metabolism and drug interactions | | | | | |
| P08246 | Neutrophil elastase | Enzyme | 0.774 | 21 | 5abw |
| **Diseases:** breast and lung cancer metastasis, cystic fibrosis airway inflammation, COVID-19–associated lung injury, atherosclerosis, Alzheimers disease with inflammatory exacerbation | | | | | |
| O14965 | Aurora kinase A | Enzyme | 0.773 | 16 | 5dn3 |
| **Diseases:** neuroendocrine prostate cancer, breast cancer, pancreatic cancer, non-small-cell lung cancer, acute myeloid leukemia | | | | | |
| P21836 | Acetylcholinesterase | Enzyme | 0.760 | 14 | 5eih |
| **Diseases:** Alzheimer's disease, Myasthenia gravis, Vascular dementia, Parkinson's disease (symptom-related), Autism spectrum disorders | | | | | |
| P39748 | Flap endonuclease 1 | Enzyme | 0.785 | 18 | 5fv7 |
| **Diseases:** oral squamous cell carcinoma, breast cancer (including paclitaxel-resistant subtype), hepatocellular carcinoma, glioma, hepatitis B virus infection (cccDNA formation) | | | | | |
| P09958 | Furin | Enzyme | 0.362 | 13 | 5mim |
| **Diseases:** SARS-CoV-2 infection, atherosclerosis, epilepsy, cancer/metastasis, MERS-CoV infection | | | | | |
| P09382 | Galectin-1 | Other | 0.643 | 23 | 5mwt |
| **Diseases:** cancer (multiple types), HIV infection, fibrotic diseases, neurodegenerative diseases, retinal diseases (nAMD, DME, RVO) | | | | | |

*Continued on next page...*

| Uniprot | Protein Name | Class | SiteScore | AA Num | PDB |
|---------|--------------|-------|-----------|--------|-----|
| P06276 | Cholinesterase | Enzyme | 0.476 | 12 | 5nn0 |
| **Diseases:** Alzheimer's disease, Myasthenia gravis (neonatal AChR-MG), Neonatal MuSK-MG, Parkinson's disease gait impairment, Dementia (including dementia with Lewy bodies) | | | | | |
| O75164 | Lysine-specific demethylase 4A | Enzyme | 0.591 | 10 | 5var |
| **Diseases:** non-small cell lung cancer, prostate cancer, glioma, hepatocellular carcinoma, neuropathic pain | | | | | |
| Q95PM0 | Cysteine protease (Fragment) | Enzyme | 0.792 | 23 | 6ex8 |
| **Diseases:** acute myeloid leukemia (AML), MLL-fusion leukemia, Wilms tumor, breast cancer (via XPO6–profilin-1–ENL axis) | | | | | |
| Q03111 | Protein ENL | Transcription / epigenetic | 0.606 | 12 | 6hpx |
| **Diseases:** acute myeloid leukemia (AML), MLL-fusion leukemia, Wilms tumor, breast cancer, kidney developmental defects associated with ENL mutations | | | | | |
| P07550 | Beta-2 adrenergic receptor | Receptor | 0.798 | 17 | 6n48 |
| **Diseases:** asthma, chronic obstructive pulmonary disease (COPD), melanoma, kidney diseases, smoking-related emphysema | | | | | |
| P10415 | Apoptosis regulator Bcl-2 | Signaling / regulatory | 0.721 | 27 | 6o0l |
| **Diseases:** dilated cardiomyopathy, triple-negative breast cancer, retinal neurodegeneration, osteosarcoma, lumbar disc degeneration | | | | | |
| P07339 | Cathepsin D | Enzyme | 0.624 | 19 | 6qbg |
| **Diseases:** breast cancer, ovarian cancer, Alzheimer's disease, triple-negative breast cancer, non-alcoholic steatohepatitis (NASH) | | | | | |
| Q8N884 | Cyclic GMP-AMP synthase | Enzyme | 0.705 | 13 | 7ftm |
| **Diseases:** non-small cell lung cancer, esophageal cancer / esophageal squamous cell carcinoma, hepatocellular carcinoma, glioma, ovarian cancer | | | | | |
| Q9Y657 | Spindlin-1 | Transcription / epigenetic | 0.784 | 15 | 7ocb |
| **Diseases:** non-small cell lung cancer, esophageal squamous cell carcinoma, hepatocellular carcinoma, glioma, ovarian cancer | | | | | |
| A5H660 | histone deacetylase | Enzyme | 0.756 | 16 | 7p2v |
| **Diseases:** peripheral T-cell lymphoma, cutaneous T-cell lymphoma, multiple myeloma, colorectal cancer (MSS/pMMR), Hodgkin lymphoma | | | | | |
| P0C6X7 | Replicase polyprotein 1ab | Other | 0.594 | 15 | 8c0g |
| **Diseases:** SARS-CoV-2 infection (COVID-19), Human coronavirus infection | | | | | |
| P61964 | WD repeat-containing protein 5 | Transcription / epigenetic | 0.748 | 15 | 8g3e |
| **Diseases:** multidrug-resistant cancer, MLL fusion leukemia, measles virus infection, nonalcoholic steatohepatitis, pulmonary hypertension | | | | | |

