# OpenReview forum: "Drugging the Undruggable: Benchmarking and Modeling Fragment-Based Screening"
_ICLR.cc/2026/Conference — ICLR 2026 Poster_

### Official Review · Reviewer_JnXH · 2025-10-20

**Soundness:** 2
**Presentation:** 2
**Contribution:** 2
**Rating:** 2
**Confidence:** 3

**Summary:**

The authors first filter undruggable pairs from PDB to construct the "FragBench" benchmark. Then build the FragCLIP system by contrastive learning.

**Strengths:**

- Undruggable target is an important research topic.
- The tri-modal contrastive learning is novel.

**Weaknesses:**

Major:

-	In Figure 1, the author only compares FragBench with DUD-E. The authors are encouraged to compare more binding datasets.

-	Some hyper-parameters are not explained. For instance, why are the site score <0.8 and LE > -0.15 used as thresholds?

-	In line 210, it said there are 1387 pock-ligand pairs; but in line 251, it said that there are only 54 undruggable targets. Are these numbers consistent? The 54 targets do not look like an adequate number for a benchmark.

-	In line 251, each target has average 183 positive fragments. It’s surprising that single target can be bind with so many fragments. Can authors provide more discussions and cases here?

-	What is the motivation to use retrieval-based methods rather than generating the fragments SMILES directly? It would also be nice to compare the generation models.

-	Are baseline methods in Table 1, e.g., DrugCLIP, finetuned on your fragment data? Otherwise, the comparison is not fair, i.e., zero-shot vs. finetuned model.

-	Data release: do authors have plans to open-source the data, which is essential for a benchmark.

Minor:

-	Some reference styles are not correct. E.g., line 52 “limited by factors like protein solubility or crystal quality Erlanson et al. (2016); Jhoti et al. (2007).”, the referred papers should in brackets.

-	In line 221, the quote marks are in wrong directions.

**Questions:**

see weaknesses.

---

> ### Author Response · Authors · 2025-11-27
>
> We sincerely thank Reviewer for the time and effort dedicated to reviewing our manuscript. We appreciate your constructive feedback, particularly regarding the evaluation benchmarks and baseline comparisons. Your suggestions have inspired us to conduct additional experiments and provide further clarifications, which we hope will strengthened the quality and completeness of our work.
>
> Our detailed responses are provided below.
>
> ---
>
> ## Response to W1 — Comparison with additional binding datasets
>
> We thank the reviewer for this suggestion. In addition to the original comparison with DUD-E, we have extended our evaluation to include two widely used virtual screening benchmarks: **DEKOIS 2.0** and **LIT-PCBA**. For both datasets, we also construct fragment-level versions by applying the same docking and PLIP-based labeling procedure used in FragBench, and we report fragment retrieval results on these new benchmarks. The dataset details are provided in the **Appendix E**, and **Figure 1** has been updated accordingly. We appreciate the reviewer’s recommendation, which has helped improve the completeness of our evaluation.
>
> ## Response to W2 — Explanation for the SiteScore and LE thresholds
>
> We thank the reviewer for highlighting that these thresholds were not sufficiently justified, and we have added an explicit explanation in the revised manuscript.
>
> - **SiteScore < 0.8.** In SiteMap, a SiteScore around **0.8** is commonly used as a practical cutoff separating more ligandable/druggable pockets from poorer sites. Following this established protocol, we set the threshold at **0.8** to intentionally focus FragBench on undruggable targets (e.g., Bio-protocol note: https://bio-protocol.org/exchange/minidetail?id=18970609&type=30; and also [1]).
>
> - **LE > −0.15.** We use ligand efficiency (LE) to filter out targets that appear easy due to unusually high per-atom binding efficiency. The cutoff **−0.15** was chosen as a **conservative** threshold based on the empirical distribution, removing only the most efficient/easy cases while retaining sufficient target coverage. This choice is consistent with standard ligand-efficiency practice in lead/fragment selection [2].
>
> [1] Stahl, Erik, et al. "Computational‐based discovery of FAK FERM domain chemical probes that inhibit HER2‐FAK cancer signaling." *Chemical Biology \& Drug Design* 95.6 (2020): 584-599.
> [2] Hopkins, Andrew L., et al. "The role of ligand efficiency metrics in drug discovery." *Nature Reviews Drug Discovery* 13.2 (2014): 105-121.
>
> ## Response to W3 — Consistency between 1,387 pocket–ligand pairs and 54 targets and is adequacy
>
> We thank the reviewer for the question. The two numbers correspond to **different stages** of our undruggable-target curation pipeline and are fully consistent.
>
> - The **1,387 pocket–ligand pairs** come from the **initial structural pre-curation stage**, where we identify all complexes in the PDB that satisfy our structural heuristics (e.g., SiteScore, LE, exclusion rules). These are *candidate instances* that may indicate undruggability.
> - The **54 undruggable targets** are the **final curated protein-level benchmark** obtained after:
>   (i) aggregating the 1,387 instances by UniProt protein,
>   (ii) performing multi-agent retrieval and evidence extraction, and
>   (iii) applying **human expert validation**.
>   Only proteins for which we found strong supporting evidence of undruggability were retained, leading to 54 high-confidence targets.
>
> Thus, the reduction from 1,387 instances to 54 targets reflects a deliberate process of consolidating structural candidates and validating them with literature-grounded evidence.
>
> Regarding the benchmark size, **54 targets falls well within the typical range of widely used structure-based screening benchmarks**. For reference: **DUD-E includes 102 targets**, **DEKOIS 2.0 includes 81**, and **LIT-PCBA includes 15**. Our benchmark therefore provides a comparable number of targets, with the distinction that all 54 are intentionally selected to represent **challenging, shallow or cryptic pockets**.

---

> ### Author Response · Authors · 2025-11-27
>
> ## Response to W4 — Why does each target have ~183 positive fragments?
>
> We thank the reviewer for raising this point. We agree that the number may look large at first glance, but it is in fact consistent with standard virtual-screening datasets and with the nature of fragment generation.
>
> First, it is common for a single target to have **hundreds of binders/actives** in widely used benchmarks; i.e., a pocket can accommodate **many chemically distinct molecules**. For example, in the original molecule-level datasets, DUD-E contains on average **221 actives per target**, and LIT-PCBA contains on average **669 actives per target**.
>
> Second, our labels are at the **fragment level**. Each active molecule can yield multiple BRICS fragments, and a subset of these fragments retains meaningful pocket interactions. Therefore, the number of positive fragments naturally scales with both the number of actives and fragmentation multiplicity, making **hundreds positives per target** a reasonable range.
>
> Finally, a “positive fragment” here does not imply a strong standalone binder; it indicates a fragment motif that forms **reproducible non-covalent interactions** with the pocket under our conservative labeling rule, providing more reliable starting points for downstream FBDD optimization.
>
> ## Response to W5 — Retrieval VS  generating fragment SMILES?
>
> We thank the reviewer for this thoughtful question. Our choice of a **retrieval-based** formulation is motivated by the fact that fragment screening in practice is typically performed over **finite, synthesizable libraries** (commercial fragment sets, DEL-based fragment selections, or internal compound collections), where the primary need is to **rank and prioritize** candidates that can be procured and tested rapidly.
>
> Retrieval and generation are also fundamentally different paradigms and are usually evaluated under different assumptions:
> - **Generative** methods can explore a much larger chemical space, but often require additional constraints (synthetic accessibility, validity, novelty, property control) and non-trivial downstream filtering, and the resulting candidates may not be readily synthesizable or purchasable.
> - **Retrieval** methods operate over a bounded candidate set, but provide actionable outputs that map directly to wet-lab validation (e.g., ordering fragments or selecting subsets for DEL/biophysical screening), which aligns closely with how fragment campaigns are executed.
>
> In this work, our goal is to establish a **standardized benchmark** and a **scalable, library-compatible** model for fragment prioritization against challenging (undruggable) pockets. FragCLIP is designed to directly accelerate these existing experimental workflows by improving enrichment within a realistic fragment pool.
>
> We agree that generative design is an exciting complementary direction and an important future avenue. However, a controlled comparison would require establishing a unified evaluation protocol (e.g., enforcing synthesizability, library constraints, novelty criteria, and downstream filtering), which is beyond the scope of this retrieval-focused benchmark. In future work, we are very interested in exploring how retrieved fragments can serve as anchors or seeds for generative models to construct full molecules—**combining the practicality of retrieval with the exploratory power of generation**.

---

> ### Author Response · Authors · 2025-11-27
>
> ## Response to W6 — Fairness of baseline (DrugCLIP)
>
> We thank the reviewer for raising this fairness concern. Our goal in the main experiments is to evaluate how well existing structure-based virtual screening methods perform on the *fragment retrieval* task. For this reason, we report DrugCLIP’s performance in its released form and use it at inference time to score fragments. This setting reflects a realistic and widely used “off-the-shelf” baseline: namely, how well a strong, state-of-the-art pocket–molecule model transfers to fragment retrieval without architectural modification or retraining.
>
> That said, we fully agree that a fair, **architecture-controlled comparison** should also include a baseline trained directly on fragment-level supervision. To address this, we added a dedicated ablation in which we remove the molecule encoder and train a **DrugCLIP-style two-tower model** on our fragment data (“**No Molecule Modality**”). This variant is **architecturally equivalent to DrugCLIP’s contrastive pocket–ligand retrieval setup but optimized specifically on the same fragment dataset as FragCLIP.**
>
>
> | Setting | AUROC | BEDROC | EF@0.5% | EF@1% | EF@2% |
> | :--- | :---: | :---: | :---: | :---: | :---: |
> | **FragCLIP** | 0.593 ± 0.018 | 0.115 ± 0.003 | 6.853 ± 0.582 | 5.797 ± 0.258 | 4.510 ± 0.163 |
> | **No Fusion Module** | 0.585 ± 0.006 | 0.105 ± 0.007 | 6.428 ± 0.241 | 5.072 ± 0.394 | 4.174 ± 0.271 |
> | **No Molecule Modality** *  | 0.584 ± 0.002 | 0.106 ± 0.006 | 6.170 ± 0.429 | 5.296 ± 0.217 | 4.049 ± 0.274 |
>
> \* **Identical to DrugCLIP trained on fragment data**
>
> ---
>
> This DrugCLIP-like baseline still underperforms the full FragCLIP model, demonstrating that FragCLIP’s improvements come not merely from finetuning on fragment data, but from the additional molecule modality and fusion module that enable richer cross-modal reasoning.
>
> In addition, we include a **No Fusion Module** variant that keeps the three encoders but disables fusion, isolating the contribution of cross-modal integration. Together, these controlled ablations provide the requested architectural head-to-head comparison and validate the necessity of both the molecule modality and the fusion module in FragCLIP.
>
> We have clarified these details in the revised manuscript and added the full controlled comparison in the **Appendix C**.
>
> ## Response to W7 — Data release plan
>
> We agree that open access to the data is essential for a useful benchmark. All datasets introduced in this work will be fully open-sourced** upon acceptance, including:
>
> - the curated FragBench targets (with metadata and evidence);
> - the fragment-level versions of DUD-E, LIT-PCBA, and DEKOIS2.0;
> - the docking poses, PLIP-derived fragment labels, and preprocessing scripts.
>
> ## Response W8 — Reference formatting and typography
>
> We thank the reviewer for pointing this out. We have corrected the citation brackets and fixed the quote mark direction in the revised manuscript.
>
> ---
>
> We hope that the additional experiments and clarifications provided above have addressed your concerns. We genuinely appreciate your valuable suggestions, as incorporating these revisions has improved the robustness of our benchmark and method.
>
> We remain fully available for further discussion and welcome any additional feedback.

---

> > ### Comment · Reviewer_JnXH · 2025-11-28
> >
> > Thank you for the response and updated results. My concerns are mostly solved, and I would like to raise my score to 6.

---

> > > ### Author Response · Authors · 2025-11-28
> > >
> > > Thank you for your response and for raising the score to 6. We are glad that our revisions have addressed most of your concerns. We truly appreciate your feedback, which has helped us improve this work. We welcome any additional feedback you may have.

---

### Official Review · Reviewer_M5Hq · 2025-10-27

**Soundness:** 2
**Presentation:** 3
**Contribution:** 3
**Rating:** 4
**Confidence:** 4

**Summary:**

This paper introduces FragBench, a new benchmark for fragment screening against undruggable proteins, and FragCLIP, a tri-modal contrastive learning model. FragCLIP jointly encodes pockets, fragments, and parent molecules into a latent space and is trained with a contrastive learning method. This paper compared FragCLIP with docking methods (Glide, Vina) and ML methods (DrugCLIP, Onion-Net, ...) and outperforms all baseline methods.

**Strengths:**

1. Despite the limited novelty of FragCLIP (similar to DrugCLIP), the motivation of this paper is pretty good. Fragment-based drug design (FBDD) is one of the most powerful tools for tackling targets previously considered "undruggable.", and I think that FragCLIP has the potential to be a useful tool for searching active fragments. Hope that you can tackle the problem of linking fragments in the future.
2. Introduce FragBench, a benchmark of curated undruggable targets with shallow or cryptic pockets.
3. Propose FragCLIP for solving the problem of drugging undruggable targets, and significantly outperforms all baseline methods on two fragment-based benchmarks (FragBench, FragBench-DUDE). FragCLIP also outperforms all baseline methods on the DUD-E benchmark.

**Weaknesses:**

1. The manuscript would benefit from an analysis of the sequence similarity between training and testing proteins. Performance is often dominated by this factor, and showing how the model performs on targets with low homology to the training set is critical for demonstrating generalization. Please consider stratifying test results by sequence identity.
2. The discussion is missing a comparison to recent high-performing methods. For example, LigUnity [1] achieves an EF 1% > 50% on DUD-E. The authors should benchmark against or, at a minimum, discuss these state-of-the-art results to properly contextualize FragCLIP's performance.
3. The evaluation should include a comparison to important recent structure-based methods like EquiScore [2] and RTMScore [3]. This is necessary to understand how FragCLIP compares to methods that explicitly model 3D interactions and geometry.


[1] Feng B, Liu Z, Li H, et al. Hierarchical affinity landscape navigation through learning a shared pocket-ligand space[J]. Patterns, 2025, 6(10).

[2] Cao D, Chen G, Jiang J, et al. Generic protein–ligand interaction scoring by integrating physical prior knowledge and data augmentation modelling[J]. Nature Machine Intelligence, 2024, 6(6): 688-700.

[3] Shen C, Zhang X, Deng Y, et al. Boosting protein–ligand binding pose prediction and virtual screening based on residue–atom distance likelihood potential and graph transformer[J]. Journal of Medicinal Chemistry, 2022, 65(15): 10691-10706.

**Questions:**

1. Why not include Dekois 2.0 and LIT-PCBA for benchmarking, just as many other works? Datasets like Dekois 2.0 and LIT-PCBA are standard in many recent works and would make the evaluation more comprehensive.
2. “Positive fragments are defined as substructures of known active ligands that make contact in a docked pose“. This is the most important assumption in this paper. However, a fragment may play roles more than binding a protein in an active ligand, e.g., acting as a linker, influencing ADMET Properties, and influencing physicochemical properties. I think maybe a more detailed discussion or analysis of this assumption will improve the quality.

---

> ### Author Response · Authors · 2025-11-27
>
> We sincerely thank Reviewer for the encouraging feedback and for recognizing the potential of FragCLIP in tackling "undruggable" targets. We particularly appreciate your constructive suggestions regarding sequence homology analysis and the inclusion of additional baselines.
> Motivated by your comments, we have conducted new experiments during the rebuttal period, including:
> **Homology analysis** (addressing W1), **Expanded benchmarking**: **LigUnity, EquiScore, and RTMScore** as baselines (addressing W2 & W3). **New datasets**: Extending our evaluation to **DEKOIS 2.0 and LIT-PCBA** (addressing Q1). All results have been updated in the draft.
>
> We hope these additions can strengthen the robustness and generalization claims of our work. Our detailed responses follow.
>
> ---
>
> ## Response to W1 — Sequence homology analysis between training and test proteins
>
> We thank the reviewer for highlighting this important point. We fully agree that sequence similarity between training and testing proteins is a critical factor influencing retrieval performance.
>
> In response to this suggestion, we conducted a rigorous homology analysis. In addition to the original setting (PDB-ID deduplication), we constructed two stricter de-homogenized datasets using **90% and 60% sequence identity thresholds**. Specifically, we removed any training protein with sequence identity exceeding these thresholds relative to any test target and retrained FragCLIP from scratch.
>
>
> **Performance under different sequence-identity thresholds**
>
> | Setting           | AUROC ± SD      | BEDROC ± SD     | EF@0.5% ± SD     | EF@1% ± SD       | EF@2% ± SD       | EF@5% ± SD       |
> |------------------|------------------|------------------|-------------------|-------------------|-------------------|-------------------|
> | **FragCLIP (100%)** | 0.637 ± 0.007    | 0.147 ± 0.006    | 10.741 ± 0.324    | 8.155 ± 0.292     | 5.737 ± 0.137     | 3.767 ± 0.077     |
> | **FragCLIP (90%)**  | 0.593 ± 0.018    | 0.115 ± 0.003    | 6.853 ± 0.582     | 5.797 ± 0.258     | 4.510 ± 0.163     | 3.000 ± 0.161     |
> | **FragCLIP (60%)**  | 0.593 ± 0.022    | 0.091 ± 0.010    | 4.843 ± 0.372     | 4.283 ± 0.750     | 3.569 ± 0.449     | 2.660 ± 0.301     |
>
>
> As expected, stricter filtering leads to a decline in performance. For instance, the AUROC decreases from 0.637 (100% setting) to 0.593 (90% setting). The most significant drop occurs when removing highly homologous proteins (100% → 90%), while further tightening the threshold to 60% results in a smaller relative decrease.
>
> The 90% de-homogenized results are now **reported in the main manuscript table**. We believe this setting provides the most practical balance between rigorously avoiding information leakage from close homologs and maintaining sufficient structural diversity for training. The full stratified results and a detailed discussion have been added to the **Appendix D**.
>
> ## Response to W2 — Comparison with LigUnity
>
> We thank the reviewer for highlighting this important baseline. We agree that comparing against state-of-the-art methods like LigUnity is essential to properly contextualize our results. In the revised manuscript, we have included **LigUnity** as a baseline. The results on the **FragBench** test set are shown below:
>
> | Method | AUROC | BEDROC | EF@0.5% | EF@1% | EF@2% | EF@5% |
> | :--- | :---: | :---: | :---: | :---: | :---: | :---: |
> | **LigUnity** | 0.505 | 0.089 | 4.262 | 3.562 | 2.933 | 2.087 |
> | **FragCLIP** | 0.593 | 0.115 | 6.853 | 5.797 | 4.510 | 3.000  |
>
> We also extended this comparison to the** DUD-E (fragment version) benchmark (results in Table 2)**.

---

> ### Author Response · Authors · 2025-11-27
>
> ## Response to W3 — Comparison with EquiScore and RTMScore
>
> We thank the reviewer for this valuable suggestion. We agree that including recent structure-based scoring models—such as **EquiScore** [2] and **RTMScore** [3]—is important for positioning FragCLIP relative to methods that explicitly model 3D geometry and residue–atom interactions.
>
> In the revised manuscript, we have **added both EquiScore and RTMScore as baselines** in our main experimental comparison. The results on the **FragBench** test set are summarized below:
>
> | Method | AUROC | BEDROC | EF@0.5% | EF@1% | EF@2% | EF@5% |
> | :--- | :---: | :---: | :---: | :---: | :---: | :---: |
> | **RTMScore** | 0.571 | 0.094 | 1.896 | 1.997 | 1.940 | 1.824 |
> | **EquiScore** | 0.581 | 0.105 | 4.039 | 3.331 | 2.638 | 2.049 |
> | **FragCLIP** | **0.593** | **0.115** | **6.853**  | **5.797** | **4.510** | **3.000** |
>
> Furthermore, to ensure a comprehensive evaluation, we also extended this comparison to the **LIT-PCBA, and Dekois 2.0** benchmarks. The detailed results for these datasets have been included in the revised manuscript (Table 2 and Appendix). We thank the reviewer again for pointing out this, which has helped improve the completeness and fairness of our evaluation.
>
>
>
> ## Response to Q1— Inclusion of DEKOIS 2.0 and LIT-PCBA
>
> We thank the reviewer for this suggestion. In response, we have added **DEKOIS 2.0** and **LIT-PCBA** datasets to our benchmark suite. For each dataset, we applied the same docking and PLIP-based labeling pipeline used in FragBench to construct **fragment-level retrieval versions**, and we now report full results for FragCLIP and all baselines on these additional benchmarks.
>
> The performance comparison on the **Dekois (fragment version)** dataset is summarized below and in manuscript. Due to space limitations, the detailed results for **LIT-PCBA** are provided in the revised manuscript (Table 2)
>
> | Method | AUROC | BEDROC | EF@0.5% | EF@1% | EF@2% | EF@5% |
> | :--- | :---: | :---: | :---: | :---: | :---: | :---: |
> | **Vina** | 0.546 | 0.053 | 3.619 | 3.022 | 2.770 | 2.330 |
> | **Glide** | 0.630 ± 0.010 | 0.065 ± 0.012 | 4.694 ± 1.964 | 4.098 ± 1.151 | 3.297 ± 0.736 | 2.597 ± 0.369 |
> | **RTMScore** | 0.506 | 0.069 | 2.153 | 1.819 | 1.569 | 1.274 |
> | **EquiScore** | 0.658 | 0.138 | 3.904 | 3.718 | 3.376 | 2.706 |
> | **DrugCLIP** | 0.640 ± 0.019 | 0.113 ± 0.002 | 8.607 ± 0.170 | 7.523 ± 0.145 | 5.937 ± 0.080 | 4.027 ± 0.175 |
> | **FragCLIP** | **0.750** ± 0.013 | **0.213** ± 0.004 | **17.963** ± 0.764 | **14.710** ± 0.447 | **10.907** ± 0.157 | **6.773** ± 0.163 |
>
> As shown, FragCLIP achieves substantial improvements over all baselines, particularly in early enrichment metrics.
>
> Dataset construction details and statistics are provided in the Appendix E, and we appreciate the reviewer’s input, which has further strengthened the breadth and completeness of our evaluation.

---

> ### Author Response · Authors · 2025-11-27
>
> ## Response to Q2 — The function of fragments
>
> We thank the reviewer for raising this important conceptual point. We fully agree that, within an active ligand, not every fragment contributes directly to binding affinity. Our framing in this work follows the classical **hit-discovery paradigm in fragment-based drug design (FBDD)**: experiments such as NMR-based screening, SPR, and DNA-encoded libraries (DELs) all begin with identifying **fragments that show measurable protein affinity**, and **afterward** consider linker design, scaffold elaboration, physicochemical tuning, and ADMET optimization during the hit-to-lead stage.
>
> Thus, for the purpose of **fragment hit identification**, direct pocket interaction is a operational definition of fragment positivity at this stage.
>
> At the same time, we fully agree with the reviewer that fragments may play roles beyond direct binding—such as stabilizing specific conformations, shaping global physicochemical properties, or tuning ADMET behavior—which become crucial in the hit-to-lead and lead-optimization stages. This is an excellent observation, and we appreciate the reviewer for bringing it to our attention. We we are considering incorporating these broader fragment functions into fragment linking and generative optimization frameworks as a highly promising direction for future work.
>
> ---
> We hope that the new experiments have addressed your concerns. We are truly grateful for your constructive feedback, which has not only helped us make this work more complete and solid but also provided inspiring insights for our future research directions.
>
> We remain fully available for further discussion and would welcome any additional feedback you may have.

---

> > ### Comment · Reviewer_M5Hq · 2025-11-28
> >
> > Thank you for your response, and I am happy to hear that my comments have provided insights for your future work. My concerns are mostly solved, and I would like to raise my score to 6. Here are two remaining questions;
> >
> > 1. Sequence similarity analysis with a strict threshold (30%) is missing. Proteins with 60% similarity are still considered highly homologous and should have high structure similarity. Experiments on this strict threshold (30%) are important to prove FragCLIP's performance on novel targets or those that were previously considered "undruggable".
> >
> > 2. Line 792: reference to the table is missing

---

> > > ### Author Response · Authors · 2025-11-28
> > >
> > > We sincerely thank the reviewer for the response and for raising the score to 6. We are glad that our previous revisions have addressed most of your concerns.
> > >
> > > Regarding your remaining questions:
> > >
> > > Sequence similarity (30%): We fully agree that testing with a strict 30% threshold is critical for demonstrating FragCLIP's generalization capabilities on truly novel targets. We are currently conducting this additional experiment and will update the results in the manuscript once ready.
> > >
> > > Missing reference: We have fixed the missing reference in the revised manuscript.
> > >
> > > Thank you again for helping us make FragBench and FragCLIP more solid.

---

> > > > ### Author Response · Authors · 2025-12-02
> > > >
> > > > ### Response to Remaining Questions Regarding Sequence Similarity (30% Threshold)
> > > >
> > > > We sincerely thank the reviewer for the encouraging feedback and for raising the score to 6. We are glad that our previous revisions addressed most of your concerns.
> > > >
> > > > Per your suggestion, we have completed the additional experiment using the strict **30% sequence identity threshold**. This ensures there is no sequence overlap $>30\%$ between the training and test sets, simulating a scenario where the targets are truly novel or historically "undruggable" with low homology to known proteins. Below are the results comparing the 30% threshold against the previously reported settings:
> > > >
> > > > | Setting | AUROC $\pm$ SD | BEDROC $\pm$ SD | EF@0.5% $\pm$ SD | EF@1% $\pm$ SD | EF@2% $\pm$ SD | EF@5% $\pm$ SD |
> > > > | :--- | :--- | :--- | :--- | :--- | :--- | :--- |
> > > > | **FragCLIP (100%)** | $0.637 \pm 0.007$ | $0.147 \pm 0.006$ | $10.741 \pm 0.324$ | $8.155 \pm 0.292$ | $5.737 \pm 0.137$ | $3.767 \pm 0.077$ |
> > > > | **FragCLIP (90%)** | $0.593 \pm 0.018$ | $0.115 \pm 0.003$ | $6.853 \pm 0.582$ | $5.797 \pm 0.258$ | $4.510 \pm 0.163$ | $3.000 \pm 0.161$ |
> > > > | **FragCLIP (60%)** | $0.593 \pm 0.022$ | $0.091 \pm 0.010$ | $4.843 \pm 0.372$ | $4.283 \pm 0.750$ | $3.569 \pm 0.449$ | $2.660 \pm 0.301$ |
> > > > | **FragCLIP (30%)** | $0.554 \pm 0.008$ | $0.079 \pm 0.006$ | $4.337 \pm 0.186$ | $3.977 \pm 0.221$ | $3.213 \pm 0.255$ | $2.270 \pm 0.053$ |
> > > >
> > > > * **100% $\to$ 90%:** The largest drop occurs here, indicating that removing highly homologous proteins (near-duplicates) has the strongest impact.
> > > > * **90% $\to$ 60%:** Further tightening the threshold produces a smaller relative decrease.
> > > > * **60% $\to$ 30%:** Notably, applying the strictest threshold of 30% results in the smallest marginal drop in performance compared to the 60% setting.
> > > >
> > > > This trend suggests that once highly similar sequences are removed, further reducing sequence identity has a **diminishing effect** on performance. These new results and discussions have been added to **Appendix D** of the revised manuscript. Thank you again for helping us strengthen the validation of FragCLIP.

---

### Official Review · Reviewer_3seo · 2025-11-01

**Soundness:** 2
**Presentation:** 2
**Contribution:** 2
**Rating:** 4
**Confidence:** 3

**Summary:**

This paper introduces a fragment-level virtual screening benchmark for hard, shallow pockets and a tri-modal model that embeds pockets, fragments, and full molecules into one space so candidates can be ranked by similarity for early enrichment. It motivates fragment retrieval as a way to reach cryptic or transient sites where docking often struggles and shows that fragment cues can also boost full-molecule screening.

**Strengths:**

1. The tackling problem is interesting and meaningful.
2. The model is explained clearly.
3. The virtual screening result seems promising

**Weaknesses:**

1.  Ground truth from docking poses is weak supervision. Labels come from Glide-generated poses plus PLIP-detected contacts, which can be wrong or biased by receptor preparation, protonation, tautomers, and grid settings. This can add false positives or false negatives and tie the dataset to a single docking engine rather than actual binding.

2. Marking a fragment as positive when “at least one atom makes one non-covalent contact” is a loose condition. A single contact can appear or disappear across docking "seeds". Fragments usually need multiple complementary contacts and burial to bind stably

3. The 54 “undruggable” targets lack basic context. The paper does not say what protein classes they are or which diseases they are linked to. Without this, readers cannot judge medical relevance or prioritize use cases, and it is unclear whether success on the benchmark would transfer to programs that help patients.

4. Table 1 does not report standard deviations, which is a problem because early enrichment (EF), especially at very small top percentages, commonly can vary a lot across random seeds, docking seeds, and splits.

5. FragCLIP follows DrugCLIP’s contrastive pocket–molecule retrieval setup and mainly adds a fragment branch plus fusion, but there is no controlled, head-to-head comparison between the two model architectures.

6. Missing citations for several previous works, such as schema-constrained decoding, validation heuristics, and baseline choices.

**Questions:**

1. Could the authors explain clearly why covalent ligand complexes, protein–ligand complexes within 6 Å of nucleic acids, and very small pockets (for example, <10 residues) were removed during curation?

2. Could the authors explain how they enforce factual accuracy in the agent–human curation loop?

---

> ### Author Response · Authors · 2025-11-27
>
> We sincerely thank Reviewer for the time and effort dedicated to reviewing our manuscript. Your insightful comments, particularly regarding label reliability and model architecture, inspired us to conduct a set of new experiments to further verify our claims. We believe these additional analyses have strengthened the solidity of our work.
>
> Our detailed responses are provided below.
>
> ## Response to W1 and W2 — Fragment label reliability and docking-based supervision
>
> We thank the reviewers for raising these important concerns. We fully agree that the accuracy of fragment-level positive/negative labels is critical, and that single-shot docking poses may be influenced by stochastic variation. Motivated by these comments, we conducted a dedicated quantitative study to evaluate how reliable docking-based fragment labels are and to refine our labeling criterion accordingly.
>
> To assess label quality rigorously, we performed a systematic evaluation on 500 PDBbind complexes. For each target, we generated fragment–pocket interaction labels from the *experimental co-crystal pose* using PLIP, which we treated as a reference ground truth. We then performed **independent random seed Glide redocking replicates** and generated labels using PLIP for each redocked pose. Comparing the docking-derived labels with the reference labels allowed us to compute fragment-level precision and recall.
>
> To examine the impact of labeling strictness—directly addressing W1 and W2—we varied two factors:
>
> 1. **Interaction threshold (addresses W2):**
>    Since a single interaction may be weak or unstable, we tested whether a fragment forms
>    **≥1 / ≥2 / ≥3 non-covalent contacts** with the pocket.
>
> 2. **Replicate-consensus threshold (addresses W1):**
>    To reduce seed-dependent variability, we required that a fragment be predicted positive in
>    **≥1 / ≥2 / 3 independent docking replicates**.
>
> This yields nine labeling configurations. The fragment-level precision, recall, and F1 scores for all settings are summarized below:
>
>
> | Interaction Requirement | Replicate Agreement | Precision | Recall |
> |-------------------------|---------------------|-----------|--------|
> | ≥1 interaction          | positive in all 3 runs | 0.934 | 0.692 |
> | ≥1 interaction          | positive in ≥2 runs    | 0.895 | 0.847 |
> | ≥1 interaction          | positive in ≥1 run     | 0.865 | 0.940 |
> | ≥2 interactions         | positive in all 3 runs | 0.894 | 0.573 |
> | ≥2 interactions         | positive in ≥2 runs    | 0.847 | 0.726 |
> | ≥2 interactions         | positive in ≥1 run     | 0.784 | 0.861 |
> | ≥3 interactions         | positive in all 3 runs | 0.883 | 0.499 |
> | ≥3 interactions         | positive in ≥2 runs    | 0.811 | 0.644 |
> | ≥3 interactions         | positive in ≥1 run     | 0.724 | 0.800 |
>
>
> Based on these results, and given that **precision is the most important factor for constructing reliable positive examples**, we adopt the conservative configuration "interaction ≥ 2 and positivity across all 3 docking replicates"
> for building the FragBench dataset. Under this setting, the positive labels achieve a **precision of 89.4%**, which provides sufficiently high confidence in the absence of large-scale experimental fragment–binding datasets.
>
> A full set of results and analyses can be found in **Appendix B**.
>
> ## Response to W3 — Context of the 54 undruggable targets
>
> We thank the reviewer for this helpful suggestion. We agree that providing biological and therapeutic context is crucial for understanding the relevance of the 54 undruggable targets. In response, we have **expanded Appendix I: FragBench Targets Details** to include, for every target, its **protein name**, **functional class**, **associated diseases**, **reference PDB structure**, and **UniProt ID**. This added information enables readers to better judge the **medical significance** of the benchmark and how model performance might translate to real-world drug discovery scenarios. Due to space limitations, we will additionally provide in the **opensource repository** a complete bibliography and more detailed annotations for all targets, including reference articles and other information. We believe these additions address the reviewer’s concern and make the benchmark substantially more **informative and actionable**.

---

> ### Author Response · Authors · 2025-11-27
>
> ## Response to W4 — Reporting standard deviations for early enrichment metrics
>
> We thank the reviewer for pointing out this important issue. In response, we have added our experiments to include **three independent runs** for FragCLIP, DrugCLIP and Glide, and we now report the mean and standard deviation of metrics in the revised result Table.
>
> Furthermore, regarding the splits, we have added in **Appendix D ** a new analysis that evaluates performance under **different levels of sequence-homology filtering**, demonstrating that our conclusions remain stable across more stringent de-homogenized splits. We hope that these additions help to alleviate the reviewer’s concern and provide greater clarity regarding the robustness of the reported results.
>
> ## Response to W5 — Head-to-head comparison with DrugCLIP-style architecture
>
> We understand the reviewer’s concern as requesting a *controlled architectural comparison*: i.e., evaluating whether FragCLIP’s gains come from the proposed **tri-modal design (pocket–fragment–molecule) and fusion**, rather than from differences in training data, label construction, or other experimental settings.
>
> To address this, we added a direct, head-to-head ablation that reduces FragCLIP to a **DrugCLIP-style two-encoder model** by removing the molecule modality and training only a pocket–fragment contrastive objective (“No Molecule Modality”). This variant is architecturally equivalent to the DrugCLIP paradigm (two-tower contrastive retrieval) but adapted to fragment retrieval data. As shown in Table, this DrugCLIP-like baseline underperforms the full FragCLIP model, demonstrating that the additional molecule branch and the fusion mechanism provide non-trivial gains beyond a standard pocket–ligand contrastive setup.
>
> In addition, we include a “No Fusion Module” variant that keeps the three encoders but disables fusion, isolating the contribution of cross-modal integration. Together, these controlled ablations provide the requested architectural head-to-head comparison and validate the necessity of both the molecule modality and the fusion module in FragCLIP.
>
> **Ablation study on model architecture**
> *Subscripts denote standard deviations across three independent runs.*
>
> | Setting | AUROC | BEDROC | EF@0.5% | EF@1% | EF@2% |
> | :--- | :---: | :---: | :---: | :---: | :---: |
> | **FragCLIP** | 0.593 ± 0.018 | 0.115 ± 0.003 | 6.853 ± 0.582 | 5.797 ± 0.258 | 4.510 ± 0.163 |
> | **No Fusion Module** | 0.585 ± 0.006 | 0.105 ± 0.007 | 6.428 ± 0.241 | 5.072 ± 0.394 | 4.174 ± 0.271 |
> | **No Molecule Modality** | 0.584 ± 0.002 | 0.106 ± 0.006 | 6.170 ± 0.429 | 5.296 ± 0.217 | 4.049 ± 0.274 |
>
> The complete quantitative ablation analysis are provided in the Appendix C.
>
> ## Response to W6 — Missing citations for related prior work
>
> We thank the reviewer for pointing this out. We have added the missing citations in the revised manuscript, and we gratefully acknowledge the important prior work that these references represent.

---

> ### Author Response · Authors · 2025-11-27
>
> ## Response to Q1 — Rationale for removing covalent complexes, nucleic-acid–adjacent pockets, and very small pockets
>
> We thank the reviewer for raising this question. Our curation choices follow standard practice in structure-based virtual screening benchmarks, and each exclusion is motivated by ensuring that the resulting pockets reflect **typical small-molecule, non-covalent drug–protein interactions**, which is the focus of FragBench.
>
> **(1) Covalent ligand complexes.**
> Covalent binders rely on *specific reactive warheads* and *irreversible chemistry*, which differ fundamentally from the non-covalent recognition dynamics that fragment-based screening aims to model. Including such complexes would mix two distinct binding modalities and introduce systematic biases unrelated to fragment–pocket complementarity.
>
> **(2) Complexes within 6 Å of nucleic acids.**
> Our benchmark specifically targets **protein pockets relevant for small-molecule drug discovery**, not RNA/DNA binding or hybrid interfaces. Pockets adjacent to nucleic acids often involve very different physicochemical environments (e.g., charge density, backbone phosphate interactions) and lead to interaction patterns that are not representative of classical medicinal-chemistry ligands. Prior virtual-screening datasets and models—such as DrugCLIP[1] —also exclude nucleic-acid–containing complexes during data preparation, for the same reason that nucleic-acid–adjacent sites follow different binding rules and would introduce heterogeneity unrelated to small-molecule drug design. Our filtering remains consistent with these widely adopted practices.
>
> [1] Gao, Bowen, et al. "Drugclip: Contrastive protein-molecule representation learning for virtual screening." Advances in Neural Information Processing Systems 36 (2023): 44595-44614.
>
> **(3) Very small pockets (<10 residues).**
> Extremely small “pockets” are often not true ligand-binding sites but rather transient crevices, crystal-packing artifacts, or positions stabilized by solvent molecules rather than biologically meaningful interactions. These sites do not support typical fragment-sized ligands and can produce misleading interaction patterns. Removing such pockets ensures that FragBench reflects realistic, chemically actionable binding sites.
>
> These filtering steps help maintain the biological relevance and interpretability of the benchmark, ensuring that the evaluated methods are judged on their ability to model meaningful small-molecule interactions with challenging protein pockets.
>
> ## Response to Q2 — Enforcing factual accuracy in the agent–human curation loop
>
> We thank the reviewer for this question. In our workflow, LLM agents are used **solely for retrieval assistance and schema-constrained extraction**, and we do not treat unconstrained model generations as factual inputs. Factual accuracy is enforced through:
>
> (1) **Grounded evidence collection.**
> All candidate targets are obtained from *retrieved* sources (DrugBank entries, PubMed abstracts, and curated biological databases). Agents operate strictly on this retrieved text.
>
> (2) **Schema-constrained structured extraction.**
> Agents extract protein metadata—such as protein name, class, associated diseases, identifiers, and undruggability rationale—into a predefined schema.
> Every field must include an explicit evidence reference (e.g., PMID or DrugBank ID), and any entry lacking verifiable evidence is discarded.
> We will also release the full evidence trail in the public repository for complete transparency.
>
> (3) **Human expert verification.**
> The final set of 54 targets was manually validated by domain experts to ensure that the extracted metadata, undruggability rationale, and all referenced information are accurate and consistent with established biological knowledge.
>
> Together, this design leverages the LLM-agent framework to perform large-scale, evidence-grounded extraction with high consistency, while human expert review serves as a final quality safeguard to ensure that all selected targets are accurate and reliable.
>
> ---
> We hope that the additional experiments and clarifications provided above have satisfactorily addressed your concerns. We genuinely appreciate your suggestions, as incorporating these analyses has significantly improved the quality and reliability of our manuscript.
>
> We remain fully available for further discussion and would welcome any additional feedback you may have.

---

### Author Response · Authors · 2025-12-03
**Summary of Rebuttal Updates and Reviewer Comment (Part 1 of 2)**

Dear AC and all reviewers,

We sincerely appreciate you stepping in to guide our paper through this transition in the review process. We also thank all reviewers for their time and efforts. During the rebuttal period, we carefully analyzed the feedback and dedicated our efforts to strengthening manuscript through new experiments. **We are pleased to receive the replies from Reviewers M5Hq and JnXH, in which they confirmed that their concerns have been mostly resolved and would like raise scores to 6.**

We understand that a new Area Chair has been assigned to our submission. To assist in your assessment, we provide a summary of the consensus and the major improvements made during the rebuttal below.

---

### Reviewers have recognized the following merits of our work:

- **Endorsement of Motivation and Problem Significance [3seo, M5Hq, JnXH]:** All reviewers recognized our motivation as "meaningful" and "important," unanimously acknowledging the critical value of addressing "undruggable" targets and introducing a dedicated benchmark for shallow and cryptic pockets.

- **Value of the Curated FragBench [M5Hq]:** FragBench provides a rigorously curated resource for identifying active fragments against challenging targets with shallow or cryptic pockets.

- **Superior Performance and Effective Design of FragCLIP [3seo, M5Hq, JnXH]:** The proposed tri-modal contrastive learning framework was recognized for its design and ability to prioritize active fragments. Reviewers highlighted that FragCLIP demonstrating strong potential for practical drug discovery applications.

---

In response to the reviewers’ constructive feedback, we have strengthened the manuscript by conducting a comprehensive expansion of experiments and analysis. To facilitate your review, all major revisions in the manuscript have been **highlighted in blue. Our revisions can be summarized in the following four key aspects:**

### 1. Concern regarding the reliability of supervision signals.

The reviewers raised concerns about the validity of our fragment labeling strategy and supervision signals. To address this, we:
- **Conducted a quantitative label reliability study[3seo]:** We performed systematic redocking experiments and interaction threshold analyses to validate our approach in **Fig. 6** and **App.B** .
- **Refined labeling criteria:** Based on the quantitative findings, we updated our fragment labeling criteria to ensure high precision **(89.4% in Tab. 4)**. We believe this establishes a **reliable fragment labeling method for benchmarking** and enable fair comparison on model performance, particularly given the current scarcity of large-scale wet-lab fragment binding data.

### 2. Suggestion regarding the comprehensiveness of the experimental evaluation.

The reviewers suggested that our evaluation could be strengthened by incorporating more diverse datasets and comparing against a wider range of baselines. To address this, we:
- **Expanded the evaluation benchmark[M5Hq]:** We added fragment-level evaluations on two additional datasets, DEKOIS 2.0 and LIT-PCBA, alongside FragBench and DUD-E.**(Fig. 1, Tab. 2)**
- **Compared with more baselines[M5Hq]:** We benchmarked LigUnity, EquiScore, and RTMScore on test datasets. **(Tab. 2)**
- **Added Standard Deviation [3seo]:** We now report the Mean and Standard Deviation in **Tab. 1 and Tab. 2** for key metrics across independent runs.

### 3. Suggestion regarding in-depth analysis of generalization and architectural contributions.
The reviewers recommended conducting further analyses to verify the source of our performance gains and to assess the model’s generalization capabilities on novel targets. To address this, we:
- **Performed strict sequence homology analysis[M5Hq]:** We evaluated under 90%, 60%, and 30% sequence identity thresholds to confirm the model's robust generalization to novel targets in **App. D and Tab. 6**.
- **Conducted detailed Head-to-Head architectural ablations[3seo,JnXH]:** We compared our method against DrugCLIP-style and no-fusion variants to isolate the specific contributions of our tri-modal design to the overall performance in **Tab. 7**.

### 4. Suggestion regarding documentation and clarity
The reviewers advised enhancing dataset documentation. To address this, we:
- **Provided detailed metadata for all 54 targets in FragBench in App. I [3seo].**
- **Provided dataset statistics[JnXH]:** We added comprehensive statistical breakdowns of the four datasets in the **App.E Tab. 8.**
- **Refined the Manuscript:** We corrected typos, updated citations and improved readability.

---

> ### Author Response · Authors · 2025-12-03
> **Summary of Rebuttal Updates and Reviewer Comment (Part 2 of 2)**
>
> In summary, the **main contribution** of our work is twofold:
>
> - FragBench: A pioneering benchmark that addresses the critical lack of large-scale wet-lab fragment data by introducing a **reliable** fragment labeling strategy specifically for undruggable targets with shallow pockets.
>
> - FragCLIP: A tri-modal model effectively bridges the representation gap between pockets, fragments, and molecules, achieving **superior performance** across challenging fragment virtual screening tasks.
>
> We are particularly gratified that Reviewers M5Hq,JnXH have confirmed their concerns were **mostly resolved and would like raise score from 4,2 to both 6**. For Reviewer 3seo, although we were unable to receive a final response due to the policy adjustment, we are confident that our comprehensive additional experiments—especially the rigorous label reliability study- have effectively addressed their concerns.
>
> Ultimately, we deeply value the insights gained from this review process and are committed to integrating these constructive suggestions into our future research endeavors. Once again, thank you for all reviewers' time and guidance.
>
> Best regards,
>
> The Authors

---

### Meta-Review · Area_Chair_q49T · 2025-12-31

**Summary:**

The paper addresses the challenge of screening fragments for undruggable protein targets. It introduces FragBench and FragCLIP, a curated benchmark constructed via a multi-agent LLM pipeline, and a tri-modal contrastive learning model integrating pocket, fragment, and whole-molecule data. Initial reviews were not positive, with primary concerns focused on the reliability of the ground truth labels, insufficient baselines, and limited dataset diversity. However, the authors executed a very good rebuttal, adding two new datasets (DEKOIS 2.0, LIT-PCBA), three new SOTA baselines (LigUnity, EquiScore, RTMScore), and a quantitative label reliability study. This comprehensive response work well and turn two reviewers to express support after rebuttal.

**Reviewer Concerns:**

## Addressed:

- Reviewer 3seo questioned the validity of using docking poses as ground truth. The authors added a systematic reliability study by redocking, demonstrating an 89.4% precision under their refined criteria.

- Reviewers M5Hq and JnXH criticized the lack of comparison against recent SOTA methods and diverse datasets. The authors added evaluations on DEKOIS 2.0 and LIT-PCBA and compared against LigUnity, EquiScore, and RTMScore, satisfying these requests.

- Reviewer M5Hq requested stricter sequence identity splits. The authors added experiments with 90, 60, 30% sequence identity thresholds

- Reviewer JnXH questioned the fairness of comparing whole-molecule trained baselines. The authors performed a "DrugCLIP-style" ablation (trained only on fragments) to prove the specific value of their architecture.

## Outstanding:
- Reviewer 3seo did not explicitly confirm the satisfaction. However, the authors' quantitative study on label reliability  directly addresses 3seo's primary technical blocker.

**Reviewer Scores:**

- Reviewer 3seo 4 ==> 4 or 6 (unchange or more likely to increase). Primary concerns were resolved.

- Reviewer M5Hq 4 ==> 6 (increase).  Concerns were resolved.

- Reviewer XoYB 2 ==> 6 (big increase). Issues were resolved.

---

### Decision · Program_Chairs · 2026-01-26

Accept (Poster)